# Open-Ocean Tides Simulated by ICON-O, Version icon-2.6.6

Jin-Song von Storch[Max Planck Institute for Meteorology], Eileen Hertwig[Deutsches Klimarechenzentrum],
Veit Lüschow[Max Planck Institute for Meteorology], Nils Brüggemann[Max Planck Institute for Meteorology],
Helmuth Haak[Max Planck Institute for Meteorology], Peter Korn[Max Planck Institute for Meteorology], and
Vikram Singh[Max Planck Institute for Meteorology]

[Max Planck Institute for Meteorology]Bundesstrasse 53, 20146 Hamburg, Germany
[Deutsches Klimarechenzentrum]Bundesstrasse 45a, 20146 Hamburg, Germany

**Correspondence:** Jin-Song von Storch (jin-song.von.storch@mpimet.mpg.de)

**Abstract.** This paper evaluates barotropic tides simulated by a newly developed multi-layer ocean general circulation - ICON-O and assesses processes and model configurations that can impact the quality of the simulated tides. Such an investigation is crucial for applications addressing internal tides that are much more difficult to evaluate than the barotropic tides. Although not specially tuned for tides and not constrained by any observations, ICON-O is capable of producing the main features of

the open-ocean barotropic tides as described by the geographical distributions of amplitude, phase, and amphidromic points. An error analysis shows however that the open-ocean tides simulated by ICON-O are less accurate than those simulated by two other OGCMs, especially when not properly adjusting the time step and the parameters used in the time stepping scheme. Based on a suite of tidal experiments we show that an increase in horizontal resolution improves only tides in shallow waters. Relevant for using ICON-O with its telescoping grid capacity, we show that spatial inhomogeneity does not deteriorate the

quality of the simulated tides. We show further that implementing a parameterization of topographic wave drag improves the quality of the simulated tides in deep ocean independent of model configuration used, whereas the implementation of SAL parameterization in a low resolution (40km) version of ICON-O degrades the quality of tides in shallow ocean. Finally, we show that the quality of tides simulated by ICON-O with low resolution (40 km) can be significantly improved by adjusting the time step or the parameters in the time stepping scheme used for obtaining the model solution.

## 1   Introduction

The development of a new climate and earth-system model ICON, carried out at the Max-Planck Institute for Meteorology in cooperation with the German meteorological service (DWD), has been advancing fast in the last years. After having documented the atmospheric and oceanic components, ICON-A and ICON-O (Giorgetta et al. (2018); Korn et al. (2022)), a standard application of the coupled ICON model in form of a set of CMIP6 simulations at a resolution typical for models participating

in the Coupled Model Intercomparison Project (CMIP) has been assessed by Jungclaus et al. (2022). The ICON-model is now used to push the frontiers of simulating the climate based on first principles by resolving the major energetic motions, including those at small scales such as the deep convection in the tropics and eddies and waves in the ocean. The activity takes the form of running storm-resolving atmosphere-alone and storm-resolving coupled atmosphere-ocean simulations (Stevens et al. (2004);

Hohenegger et al. (2022)), a development referred to as ICON-Sapphire. Here "storm-resolving" refers to resolutions of a few kilometers (well-below ten kilometers). This new development however requires further efforts. Some of them are needed to better represent potentially important processes which still elude a physical description, such as the mixing processes that occur at hectometer and finer scales. Others are needed in order to better represent phenomena resolved by kilometer-scale models. One such phenomenon is internal tides that are generated as barotropic tides flow over topographic features, such as underwater ridges and sea mounts. To ensure a realistic representation of internal tides, one needs to make sure that the model used is capable to realistically simulate the barotropic tides. This paper evaluates barotropic tides simulated by ICON-O, and assesses processes and model configurations that can impact the quality of the simulated tides.

Internal tides are internal waves at the tidal frequencies. As one of the most energetic components of the ocean internal wave field, internal tides play an important role for the ocean general circulation. It is thought that the breaking of internal waves provides the power needed to maintain the oceanic overturning circulation (Munk and Wunsch (1998); Wunsch and Ferrari (2004); Ferrari and Wunsch (2009)). The storm-resolving models provide a new tool for studying internal tides. In fact the collaborative research center TRR181 founded by the German science foundation made a proposal to advance the investigation of internal tides and their interactions with mesoscale eddies using a storm-resolving ICON-O. To further enhance the horizontal resolution in specific regions, a telescoping configuration is developed (Korn et al. (2022)). The target configuration has a fine resolution below one kilometer, down to about 600 meters, within a focus domain of about several thousand kilometers. With such a fine resolution, the proposal aims to simulate a larger range of vertical modes associated with internal tides.

A prerequisite for a realistic simulation of internal tides is a realistic simulation of barotropic tides. Modelling barotropic tides has a long history. As reviewed by Stammer et al. (2014), the modern tidal models, that are barotropic models tightly constrained by satellite altimeter data, produce much more accurate tides than hydrodynamic unconstrained tidal models. Typically, the differences between the observed bottom pressure recorders and the unconstrained models are about a factor of 10 larger than corresponding differences for the modern tidal models that assimilate altimetry. This large difference reflects the overall difficulty in accurately simulating tides without constraining the model with observations, as it is usually the case when developing models for climate purposes. Generally, modelling tides using global ocean general circulation models (OGCM) is much less matured than modelling tides using barotropic tidal models. To our knowledge, modelling tides using multi-layer OGCMs (for the purpose of studying internal tides) have only been carried out in HYCOM simulations using the HYbrid Coordinate Ocean Model (Arbic et al. (2010); Arbic et al. (2012)) and in STORMTIDE / STORMTIDE2 using the Max-Planck Ocean Model (MPIOM) (Mueller et al. (2012); Li and von Storch (2020)). For the new ocean model ICON-O, only the coastal M2 tide has been evaluated (Logemann et al. (2021)). No evaluation has been performed concerning the open-ocean tides. It is also not clear whether and how various model configurations impact the quantity of the open-ocean tides simulated by ICON-O. Addressing these issues is an important step towards a storm-resolving OGCM capable of simulating both mesoscale and sub-mesoascale eddies and a large range of modes of internal tides in general, and towards a successful implementation of the numerical experiments planned by TRR181 in particular.

We note that accurately simulating tides can be beneficial for a large range of applications. Apart from studying internal tides, an important application is to provide tidal corrections to various measurements so that small non-tidal signals can be identified and investigated (Knudsen and Andersen (2002)). It is conceivable that different applications require different degrees of accuracy of the simulated tides. In particular, tides for the purpose of studying overall behavior of internal tides may need not be simulated as accurately as tides used for providing tidal corrections. Adequately addressing this difference is beyond the scope of this paper. We will only briefly discuss this issue at the end of paper.

This paper provides a first systematic evaluation of the tides simulated by ICON-O. Such an evaluation becomes possible, after the implementation of the lunisolar tidal potential into ICON-O by Logemann et al. (2021). Different from the study by Logemann et al. (2021), which focuses on coastal tides simulated by the ICON-O with highly irregular meshes suitable to coastal studies, this paper focus on open-ocean tides for studying internal tides in the ocean interior.

After describing in Section 2 the suite of tidal experiments performed and the analysis methods used, we assess the quality of the tides simulated by a standard configuration of ICON-O in Section 3. We then evaluate in Section 4 the effect of the horizontal resolution, the effect of the horizontal in-homogeneity of the grid, and the effect of the vertical coordinate on the quality of the tides. Section 5 describes the effects of parameterizations of topographic wave drag and the self-attraction and loading on the quality of the simulated tides. A summary and discussions are given in the final section.

## 2 Experiments and error analysis methods

### 2.1 The suite of tidal experiments

The tidal experiments are performed with ICON-O - an ocean circulation model based on primitive equations. The momentum equation includes as the luni-solar tidal forcing the horizontal component of the difference between the gravitational force due to the celestial body (moon and sun) and the centrifugal force caused by the earth's rotation around the centre of mass of the earth/celestial body system (Logemann et al. (2021)). The novelty of the ICON-O numerics is the use of specific recon-structions that are required to combine scalar and velocity fields for flux calculation (Korn (2017)). These reconstructions are compatible with a discrete scalar product for the state space of ICON-O consisting of horizontal velocity, potential tempera-ture, salinity and the surface elevation. This allows writing the discrete primitive equations in a discrete weak/variational form, a necessary condition to derive discrete conservation principles. The design of ICON-O has been centred on these discrete conservation principles.

The horizontal grid of ICON is created by recursively dividing the original 20 triangles of the icosahedron, via bisecting the edges, which results in a so-called R2Bn grid with $n$ indicating the number of subdivisions used (Giorgetta et al. (2018)). The horizontal resolution is estimated as the square root of the average triangular area, which may overestimate the true effective resolution (Danilov (2022)). The bottom topography is interpolated from SRTM30 (Farr et al. (2007)). The vertical coordinate-axis of ICON-O is given either in the z-coordinate, or in the z*-coordinate. When using the z* coordinate, the primitive equations are solved following Adcroft et al. (2004) on a virtual grid with vertical coordinate being scaled in proportion with the sea-surface elevation. The formulation in z*-coordinate ensures that the structure preserving machinery developed for the

z-coordinate can be adopted so that tracer content, variance, and energy are conserved. Both z- and z*-coordinate employ total 128 layers, with a vertical spacing that is smaller than 10 m (apart the first layer in z-coordinate) in the upper 140 m and smaller than 100 m in the upper 3450 m, and identical to 200m below 4544 m.

In all numerical experiments shown here we employ as horizontal friction a grid-size dependent biharmonic viscosity. The vertical mixing is parameterized using the TKE scheme following Gaspar et al. (1990). The effect of meso-scale eddies is parameterized following Griffies (1998), which is switched on for the simulation at a resolution of about 40 km (R2B6, defined below), but switched off otherwise.

The suite of tidal experiments (Tab.1) are designed to address the following three questions. First, how accurate are the tides simulated by a standard configuration of ICON-O? We consider ICON-O configured with a R2B6 grid with a horizontal resolution of about about 40 km in z-coordinate as our standard configuration. This choice is made not only because that ICON-O in R2B6 resolution is normally used for climate simulations (Jungclaus et al. (2022)). More importantly, a resolution of 40km allows by and large an adequate representation of topographic features characterized by underwater ridges, sea mounts, passages, straits. This could be (at least partially) the reason that the modern data-constrained tidal models can reach already a high accuracy at a resolution of about 0.5° (Stammer et al. (2014)).

Secondly, we address the question of whether and how different model configurations, set by horizontal resolution, spatial inhomogeneity of the grid, and different vertical coordinates (z-coordinate versus z*-coordinate), affect the quality or accuracy of the simulated tides. We do so by performing tidal experiments in which one of the three aspects - horizontal resolution, horizontal inhomogeneity, vertical coordinate - is modified, while keeping all other aspects of ICON-O almost unchanged.

Generally, it is expected that increasing horizontal resolution could improve the accuracy of the simulated tides. Does this improvement concern only the tides in the coastal region or do we see also improvement for the open-ocean tides? We assess the effect of horizontal resolution by increasing the horizontal resolution from the standard R2B6 configuration (about 40 km) to R2B8 (about 10 km).

A strong spatial inhomogeneity is introduced by a telescoping grid, which can affect the quality of the simulated tides. When wobbling from a region with higher resolution to a region with lower resolution, tides can be deformed and deteriorated. We assess the effect of spatial inhomogeneity by comparing the tides simulated by ICON-O on a telescoping grid with the tides simulated by ICON-O on the standard R2B6 grid. We use a "basecamp-telescope" (BCT) grid shown in Fig.1, whose focus domain lies in the South Atlantic over the Walvis Ridge and has a resolution of about 8 km. The focus domain was dictated by the observational campaign carried out for TRR181. Outside the focus domain of about several thousand kilometers, the resolution decreases gradually to the coarsest value of about 80 km. Even though much coarser than the telescoping grid that will be used in the final runs proposed by TRR181, the BCT-grid has roughly the same degree of inhomogeneity, i.e. roughly the same ratio of the smallest to the largest grid size.

In addition to horizontal resolution and spatial inhomogeneity, we consider also the effect of vertical coordinates. The experiments proposed by TRR181 will be done with ICON-O in the z*-coordinate with a thin surface layer to better resolve sub-mesoscales near the surface. When using standard z-coordinate, the first layer must be sufficiently thick to "absorb" both a time-varying sea-ice growth and a time-varying sea surface elevation, which could be large locally in the presence of tides.

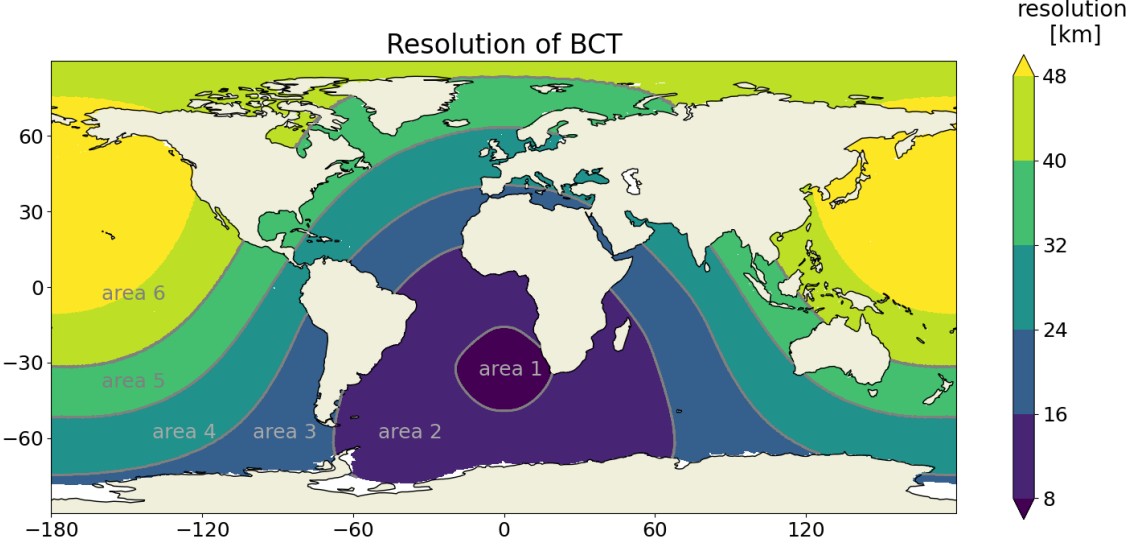

**Figure 1.** Resolution of the telescope grid BCT in km.

In fact, even with a surface layer of 11 m, the model becomes unstable when switching on the tides. In order to complete the simulations with tides, we had to further increase the thickness of the first layer from 11 meters to 14 meters (indicated in the fifth column of Tab.1). When switching to z*-coordinate, we employ a surface layer of 2 m, which is needed for a better representation of near-surface sub-mesoscale motions. Employing a thin surface layer leads to a change in the distribution of the 128 vertical layers. This change, only noticeable in the upper ocean, is illustrated in Fig.2. Generally, we do not expect big changes in the quality of the simulated tides when transforming from z- to z*-coordinate. We rather consider the result obtained with a R2B6 ICON-O in z*-coordinate as a further technical check.

Finally, we address also the question of whether and to what extent parameterizations of two secondary tidal processes affect the quality of the simulated tides. One of these process is Self-Attraction and Loading (SAL), and the other is the Topographic-Wave-Drag (TWD), which describes the transfer of the tidal energy to the internal tide energy. While the SAL effects have been mostly included in the barotropic tidal models (Ray (1998); Accad et al. (1978)), attempt is also made to include the SAL effects in an OGCM (Shihora et al. (2022)).

The suite of tidal experiments used to address the above questions are listed in Tab.1. They can be classified into three groups. The first one (blue) comprise spin-up runs (Hertwig et al. (2021)). Since we aim to simulate tides together with the oceanic circulation, and since a change in the model configuration, e.g. a change in the horizontal resolution, can make the model to drift to a somewhat different state, all tidal experiments start from a state at the end of a respective spin-up run. A spin-up run starts from the temperature and salinity fields interpolated from the Polar Science Center Hydrographic Climatology PHC2

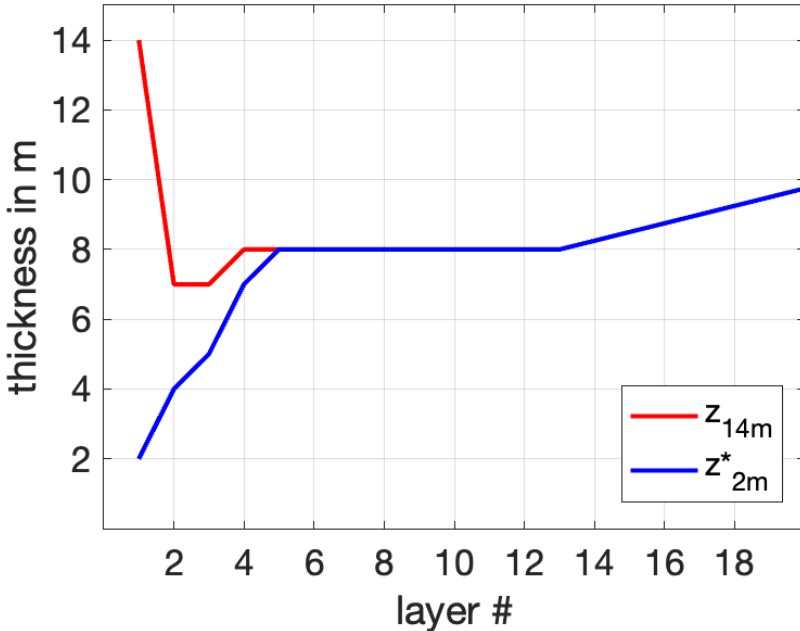

**Figure 2.** Layer thickness as a function of layer number used in the z- (red) and z*-coordinate (blue) configuration.

(Steele et al. (2001)) from rest and is forced by the OMIP forcing Roeske (2006) without tides. After 100 years, the climate has spun up reasonably. Fig.3 shows for example that the AMOC at 26°N is by and large stabilized in all spin-up runs in the last decades.

The second (bold) and third (non-bold) group are the actual tide experiments (Hertwig et al. (2021, 2022)). The experiments in bold are used to assess the effects of horizontal resolution, vertical coordinate, and the spatial inhomogeneity on the simulated tides. The other experiments (non-bold) are used to assess the effect of SAL and TWD in different model configurations. All tidal experiments start from year 97 of the respective spinup-run and are forced by both the OMIP forcing and by the tidal potential for three years. For the last of the three years, we output the sea surface elevation at a hourly frequency (Hertwig et al. (2021)). The analysis will be carried out for the one-year hourly output of each experiment.

## 2.2 Error analysis

To evaluate the quality of the simulated tides, we use mainly the tidal model product TPXO9, the most recent version of a global barotropic model of ocean tides, which best-fits (in a least-squares sense) the Laplace Tidal Equations and altimetry data and will hereafter be considered as our proxy observation. More details about TPXO can be found in Egbert et al. (1994) and Egbert and Erofeeva (2002). The tides are harmonic analyzed by the least-square-fitting method of Foreman et al. (2009). From this harmonic analysis the amplitudes and phases of the eight major diurnal and semi-diurnal tidal constituents (M2, S2, N2, K2, K1, O1, P1, and Q1) are derived. The amplitudes and phases of the simulated tides are compared with those obtained from our proxy observation, TPXO9. For the comparison, the model grid is interpolated into the TPXO9 grid.

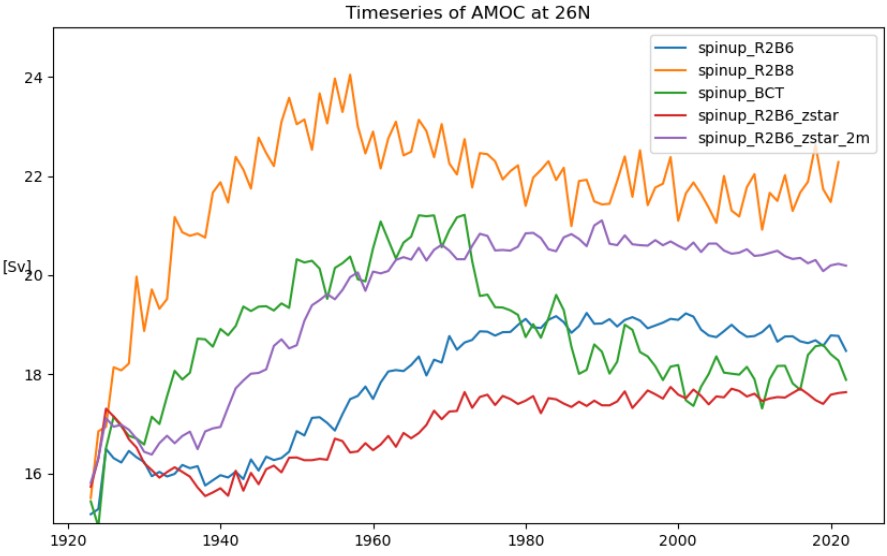

**Figure 3.** Timeseries of Atlantic Meridional Overturning Circulation defined as the maximum of zonally averaged streamfunction at 26°N, derived from the four spinup experiments listed in Tab.1.

Following Arbic et al. (2004) and Stammer et al. (2014), we quantify the fidelity of the simulated $k$-th tidal constituent at a grid point in terms of the total error defined as the squared difference $d_k^2$ between the simulated and observed tides at that grid point averaged over a tidal period

$$
\begin{aligned}
d_k^2 &= \ < (A_o \cos(\omega t - \phi_o) - A_m \cos(\omega t - \phi_m))^2 > \\
&= \ \frac{1}{2}(A_o^2 + A_m^2) - A_o A_m \cos(\phi_o - \phi_m) \\
&= \ \frac{1}{2}(A_o - A_m)^2 + A_o A_m \big(1 - cos(\phi_o - \phi_m)\big),
\end{aligned}
\tag{1}
$$

where $A_o$ and $A_m$ are the amplitudes, $\phi_o$ and $\phi_m$ the phases of the $k$-th tidal constituent identified from TPXO9 (denoted by the subscript $_o$ for "observations") and from ICON-O simulation (denoted by $_m$ for "model"), and $< \cdot >$ denotes an average over the respective tidal period.

It is obvious from Eq.(1) that the total error as measured by $d_k^2$ can be decomposed into two parts

$$
d_k^2 = d_{k,AM}^2 + d_{k,PH}^2,
\tag{2}
$$

with

$$
d_{k,AM}^2 = \frac{1}{2}(A_o - A_m)^2,
\tag{3}
$$

| Name | resolution | years | starting from | vertical coordinate (top level height) | SAL | TWD |
|------|-----------|-------|---------------|----------------------------------------|-----|-----|
| *spinup_R2B6* | R2B6 L128 | 100 | PHC3 | z (11m) | no | no |
| **R2B6** | **R2B6 L128** | **3** | ***spinup_R2B6*** | **z (14m)** | **no** | **no** |
| R2B6_SAL | R2B6 L128 | 3 | spinup_R2B6 | z (14m) | yes | no |
| R2B6_TWD | R2B6 L128 | 3 | spinup_R2B6 | z (14m) | no | yes |
| R2B6_TWD_SAL | R2B6 L128 | 3 | spinup_R2B6 | z (14m) | yes | yes |
| *spinup_R2B8* | R2B8 L128 | 100 | PHC3 | z (11m) | no | no |
| **R2B8** | **R2B8 L128** | **3** | ***spinup_R2B8*** | **z (14m)** | **no** | **no** |
| R2B8_SAL | R2B8 L128 | 3 | spinup_R2B8 | z (14m) | yes | no |
| R2B8_TWD | R2B8 L128 | 3 | spinup_R2B8 | z (14m) | no | yes |
| R2B8_TWD_SAL | R2B8 L128 | 3 | spinup_R2B8 | z (14m) | yes | yes |
| *spinup_BCT* | BCT L128 | 100 | PHC3 | z (11m) | no | no |
| **BCT** | **BCT L128** | **3** | ***spinup_BCT*** | **z (14m)** | **no** | **no** |
| BCT_SAL | BCT L128 | 3 | spinup_BCT | z (14m) | yes | no |
| BCT_TWD | BCT L128 | 3 | spinup_BCT | z (14m) | no | yes |
| BCT_TWD_SAL | BCT L128 | 3 | spinup_BCT | z (14m) | yes | yes |
| *spinup_R2B6_zstar_2m* | R2B6 L128 | 100 | PHC3 | z* (2m) | no | no |
| **R2B6_zstar_2m** | **R2B6 L128** | **3** | ***spinup_R2B6_zstar_2m*** | **z* (2m)** | **no** | **no** |

**Table 1.** Names and further specifications of the suite of simulations considered in this paper, consisting of spinup simulations (blue), simulations used to quantify the effect of resolution, spatial inhomogeneity and vertical coordinate (bold), and simulations using to assess the effects of SAL and TDB on the quality of tides in different configurations (non-bold). R2B6 denotes the use of a resolution of about 40 kilometers, R2B8 the use of a resolution of about 10 kilometers, and BCT stands for Base-Camp-Telescope. L128 means 128 unequally spaced vertical levels. PHC3 refers to Polar Science Center Hydrographic Climatology (Steele et al. (2001)). Only the height of the first levels varies between the runs and is given together with the vertical coordinate (z or z*). SAL stands for the parameterization for Self-Attraction and Loading and TWD for the parameterization for Topographic Wave Drag.

and

$$d_{k,PH}^2 = A_m A_o (1 - \cos(\phi_o - \phi_m)). \tag{4}$$

$d_{d,AM}^2$ measures the errors arising solely from model's inability to correctly simulate the amplitude of a tide, and can be obtained by letting in Eq.(1) $\phi_m = \phi_o$. $d_{d,PH}^2$ measures the errors arsing solely from model's inability to correctly simulated the phase of a tide, and can be obtained by letting in Eq.(1) $A_m = A_o$.

The errors described by the local measures $d_k^2$, $d_{k,AM}^2$, and $d_{k,PH}^2$ can be further summarized by averaging over an area (or over a set of collected grid points). The results are denoted by

$$D_k^2 = \overline{(d_k^2)}, \ \ D_{k,AM}^2 = \overline{(d_{k,AM}^2)}, \ \ D_{k,PH}^2 = \overline{(d_{k,PH}^2)}, \tag{5}$$

where $\overline{(\cdot)}$ indicates an average over grid points.

To quantify the overall performance of the $M = 8$ tidal constituents listed above, we follow Arbic et al. (2004) and consider two additional measures. One measures the combined error for all eight tidal constituents, referred to as $RSS$ (root square

sum) of the error,

$$RSS^2 = \sum_{k=1}^{M} D_k^2. \tag{6}$$

The other is the skill score

$$E = 100\% \times \left(1 - \frac{RSS^2}{\sum\limits_{k=1}^{M} S_k^2}\right), \tag{7}$$

defined using the total error measured by $RSS^2$ relative to the sum over the observed signals $S_k$

$$S_k^2 = \frac{\overline{A_o^2}}{2}, \tag{8}$$

where $A_o$ denotes the amplitude of the $k$-th tidal constituent obtained from TPXO. $E$ equals 100%, if all eight tidal constituents are perfectly simulated by the model. Both $RSS$ defined in Eq.(6) and the skill $E$ defined in Eq.(7) can also be calculated for errors with respect to amplitude or phase by replacing $d_k$ in RSS by $d_{k,AM}$ and $d_{k,PH}$ respectively (i.e. $RSS_{AM}$ and $RSS_{PH}$ $E_{AM}$, and $E_{PH}$ respectively).

## 3 Tides simulated by ICON-O in the standard R2B6 configuration

Generally, there is an overall good agreement between tides simulated by ICON-O in its standard R2B6 configuration and the tides in TPXO9. For the $M_2$ tide - the strongest tidal constituent in the ocean, the geographical distributions of amplitude (colors) and phase (black lines), including the amphidromic points, obtained from ICON-O (top panel of Fig.4) are by and large comparable to those obtained from TPXO9 (bottom panel of Fig.4). Similar comparability between R2B6 and TPXO9 is found for the $K_1$ tides - the strongest diurnal tide (Fig.5). The most obvious difference is the too strong amplitude in R2B6, especially for the $M_2$ tide in the Atlantic. Some noticeable phase differences are found for the $K_1$ tide in the Southern Ocean. Another difference concerns the tiny "wiggles" of the lines of constant phase, which is found in R2B6 (especially for the $K_1$ tide in the Pacific in Fig.5) , but clearly absent in TPXO9. Arbic et al. (2012) found similar wiggles in the phase lines of $M_2$ tide simulated by HYCOM and attributed them to the simulated $M_2$ internal tides. The wave length of the $K_1$ internal tide, about 200 - 250 km equatorward of the critical latitudes (Li et al. (2017)), is longer than that of $M_2$ internal tide, which is maximal about 150 km (Li et al. (2015)). Since waves with longer wavelength can be more easily simulated, the "wiggles" are more pronounced for the phase lines of the $K_1$ tide than those of the $M_2$ tide.

To further quantify the errors against TPXO9, we show in the top panels of Fig.6 - Fig.8 maps of the differences $d_k$, $d_{k,AM}$ and $d_{k,PH}$ for $k =$$M_2$ in R2B6. The largest errors in $M_2$ tide are found off the coast of Iberian Peninsula, in the Labrador Sea, and west of Central America (top panel in Fig.6), where the $M_2$ tide has its largest amplitude in TPXO9 (color shadings in the bottom panel of Fig.4). It seems that there is a correlation between the magnitude of the errors and the strength of the signal. The top panels in Fig.7 and Fig.8 show that these errors arise from the inaccuracy in simulating both the amplitude and the

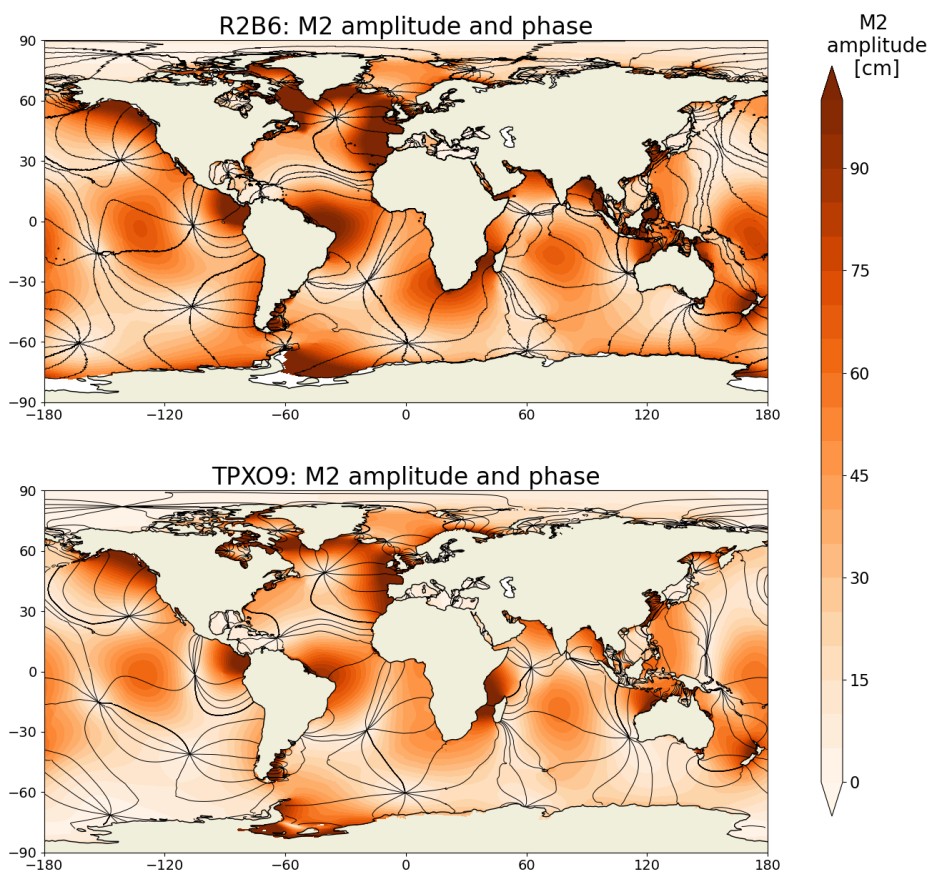

**Figure 4.** Amplitude (in cm) and phase of the $M_2$ tide simulated by ICON-O in the standard R2B6 configuration (top) and derived from TPXO9 (bottom).

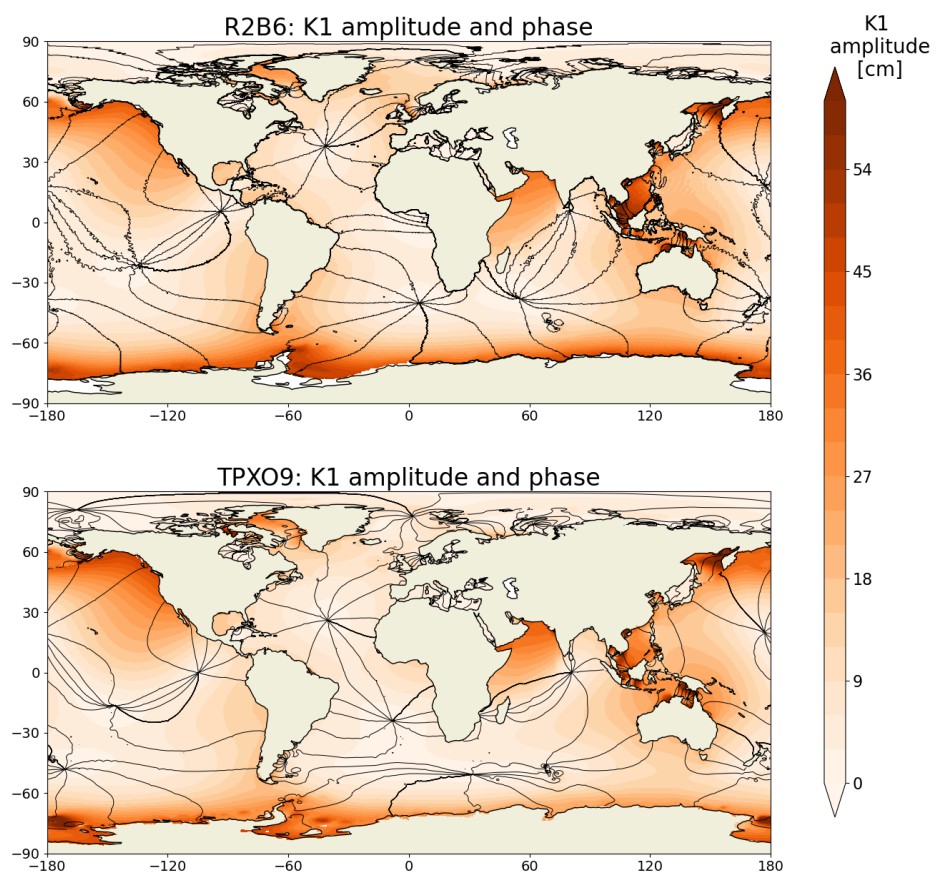

**Figure 5.** Same as Fig.4, but for the K$_1$ tide.

phase of the tide, with the contribution from amplitude error being larger than that from phase error in the North Atlantic, but vice versa for the errors west of Central America.

Large errors are also found in the Southern Ocean along the cost of West Antarctica west of the Drake Passage, where $M_2$ tide has small amplitudes in TPXO9 (color shadings in the bottom panel of Fig.4). The study by Pal et al. (2023) suggest that these errors can be induced by ignoring Antarctic ice shelf cavities in ICON-O. The top panels in Fig.7 and Fig.8 show further that these errors result mainly from amplitude errors.

Tab.2 summaries the signals, the errors measured by $D_k$, $D_{k,AM}$, $D_{k,PH}$ for each constituent, and the overall skill $E$ averaged for all constituents (last column). To focus on the open-ocean tides, all numbers are obtained by averaging over grid points deeper than 1000 meters. We see that the signals of the R2B6 tides (second row of numbers) are stronger than those of the TPXO9 tides (first row). For the four semi-diurnal tides, the largest value of $D_k$, which amounts 14.25 cm, is found for the $M_2$ tide (third row). This error is related to a strong overestimation of amplitude: While the TPXO9 $M_2$ signal is about 24.5 cm, the R2B6 $M_2$ signal is 31.67 cm. This overestimation in amplitude is reflected in the larger value of $D_{M_2,AM}$ (4th row) than the value of $D_{M_2,PH}$ (5th row). For the four diurnal tides, the largest value of $D_k$ of 5.19 cm is found for the $O_1$ tide, the second strongest diurnal tide. As for the $M_2$ tide, the amplitude seems to contribute more to the total error than the phase.

For the $M_2$ tide, the averaged error $D_k$ of 14.25 cm is clearly larger than those produced by all the unconstrained barotropic and baroclinic hydrodynamic models discussed in Stammer et al. (2014). Their Tab.12 shows averaged errors mostly smaller than 10 cm. When averaged over all eight constituents, the overall skill $E$ is only about 63%. Ignoring the errors in amplitude or in phase increases the skill to about 81%.

The large averaged errors and the low skill suggest that the tides in ICON-O are less accurate than those found in the other unconstrained models. To further confirm this impression, the tides simulated by the different models must be evaluated against the same observational evidence. For the two OGCMs considered in Stammer et al. (2014), HYCOM with a horizontal resolution of $1/12.5°$ and MPIOM with a horizontal resolution of $1/10°$, error analyses against the 102 pelagic tide measurements Shum et al. (1997) are available. We hence repeated the analysis for R2B6, but now against the pelagic data. We obtained for the R2B6 tides the $E$-value of 77.4%, which is clearly lower that 89.02 % in STORMTIDE2 (Li and von Storch (2020)), 92.8% in STORMTIDE (Mueller et al. (2012)), and 92.6 % in HYCOM (Arbic et al. (2010)). Although the value of 77.4% averaged over pelagic stations is somewhat larger than 63.38% averaged over grid points deeper than 1000 meters (last column in Tab.2), the result clearly shows that the ICON-O in its standard R2B6 configuration is less skillful in simulating tides than the $1/12.5°$ HYCOM and $1/10°$ MPIOM. It is tempting to attribute the low skill to the lower horizontal resolution of about 40 km relative to the resolution of about 10 and 8 km in STORMTIDE / STORMTIDE2 and the HYCOM simulation. As to be discussed in the next section, increasing horizontal resolution by using a R2B8 grid improves some aspects, but not the skill in the deep ocean.

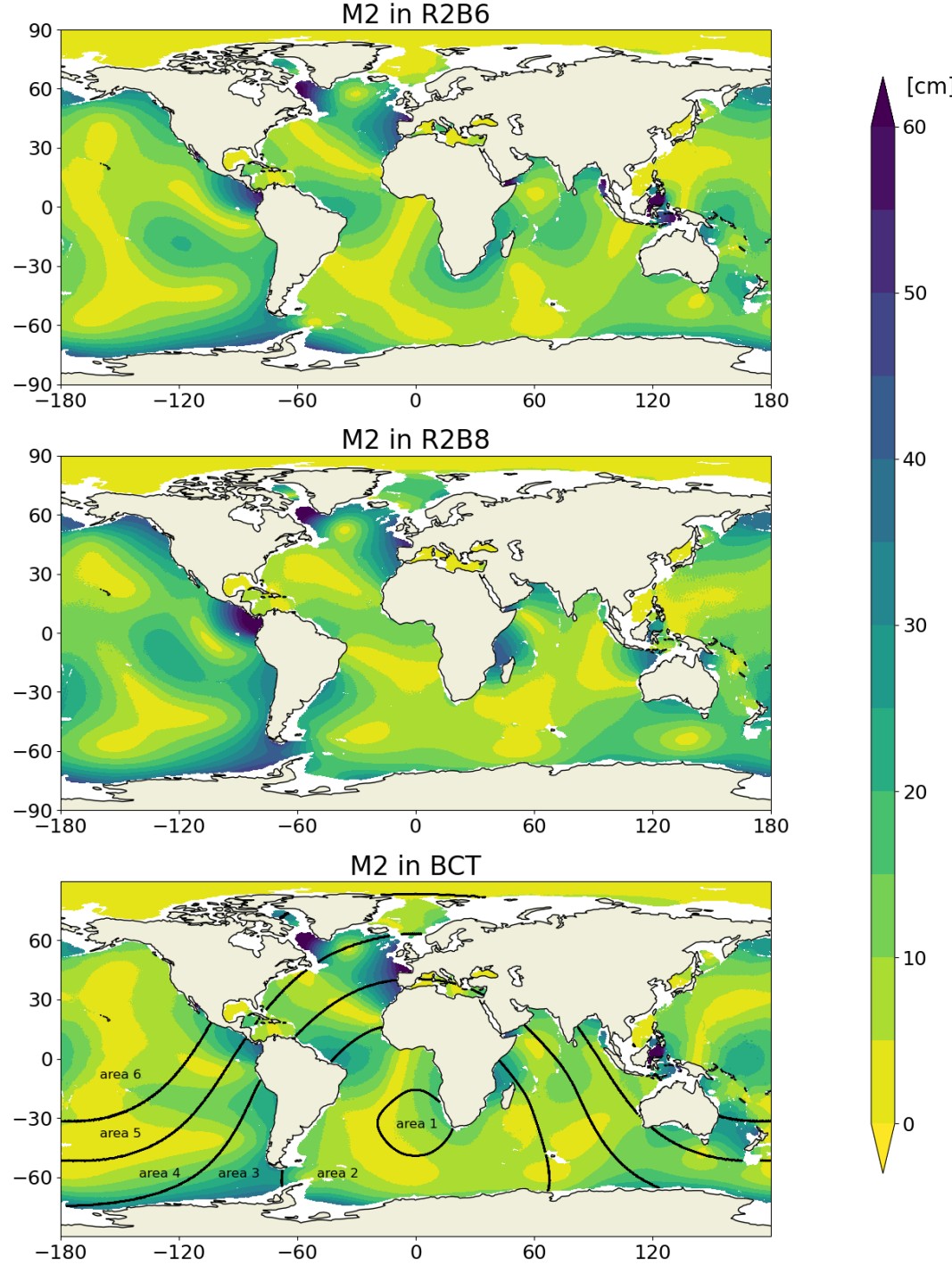

**Figure 6.** Difference $d_k$ resulting from the total error of the $M_2$ tide in ICON-O in the R2B6 (top), R2B8 (middle) and BCT configuration (bottom). $d_k$ is defined in Eq.(1) and evaluated against TPXO9. Unit is cm.

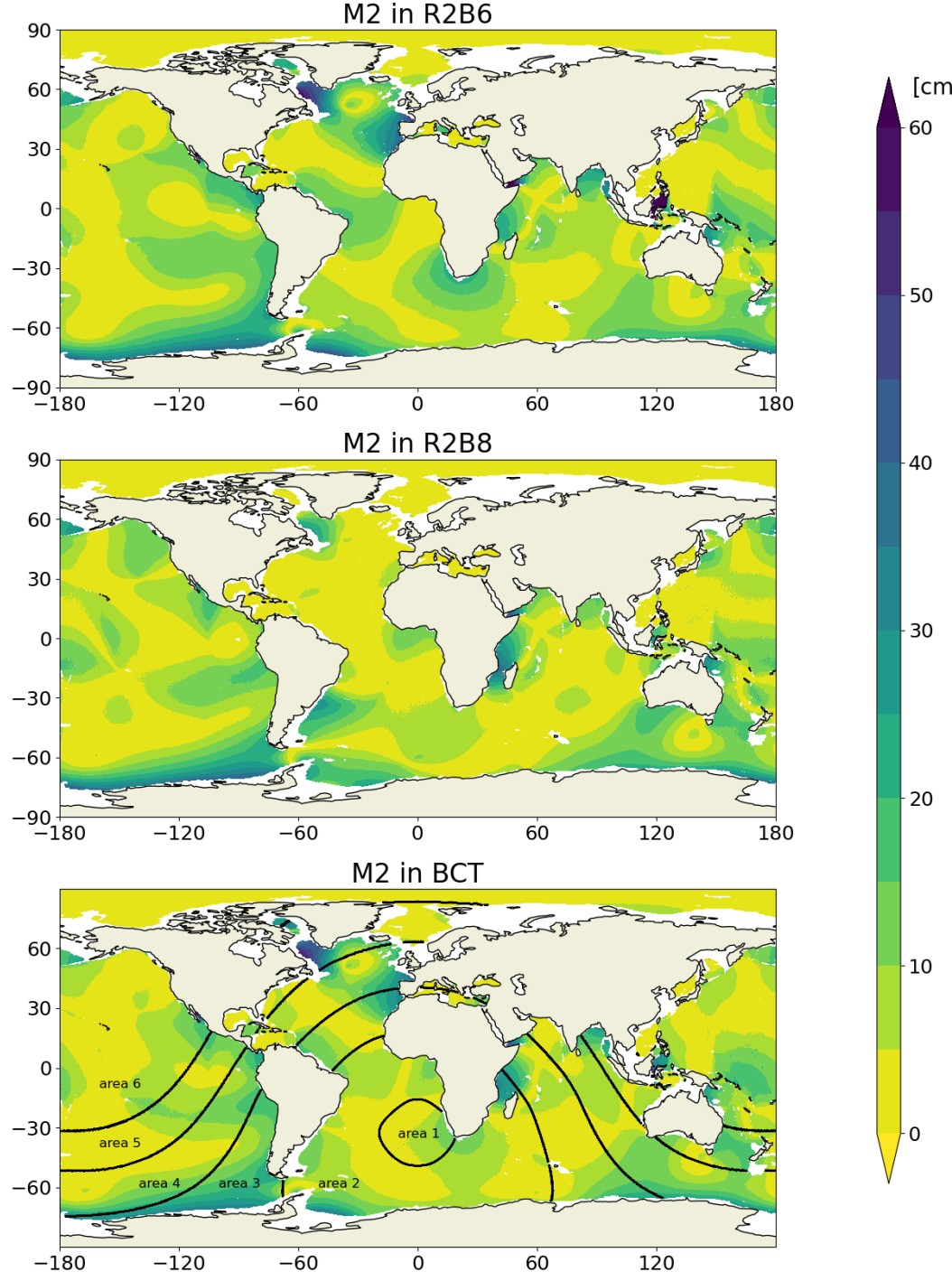

**Figure 7.** Same as Fig.4 but for the difference $d_{k,AM}$ resulting from errors in amplitude only.

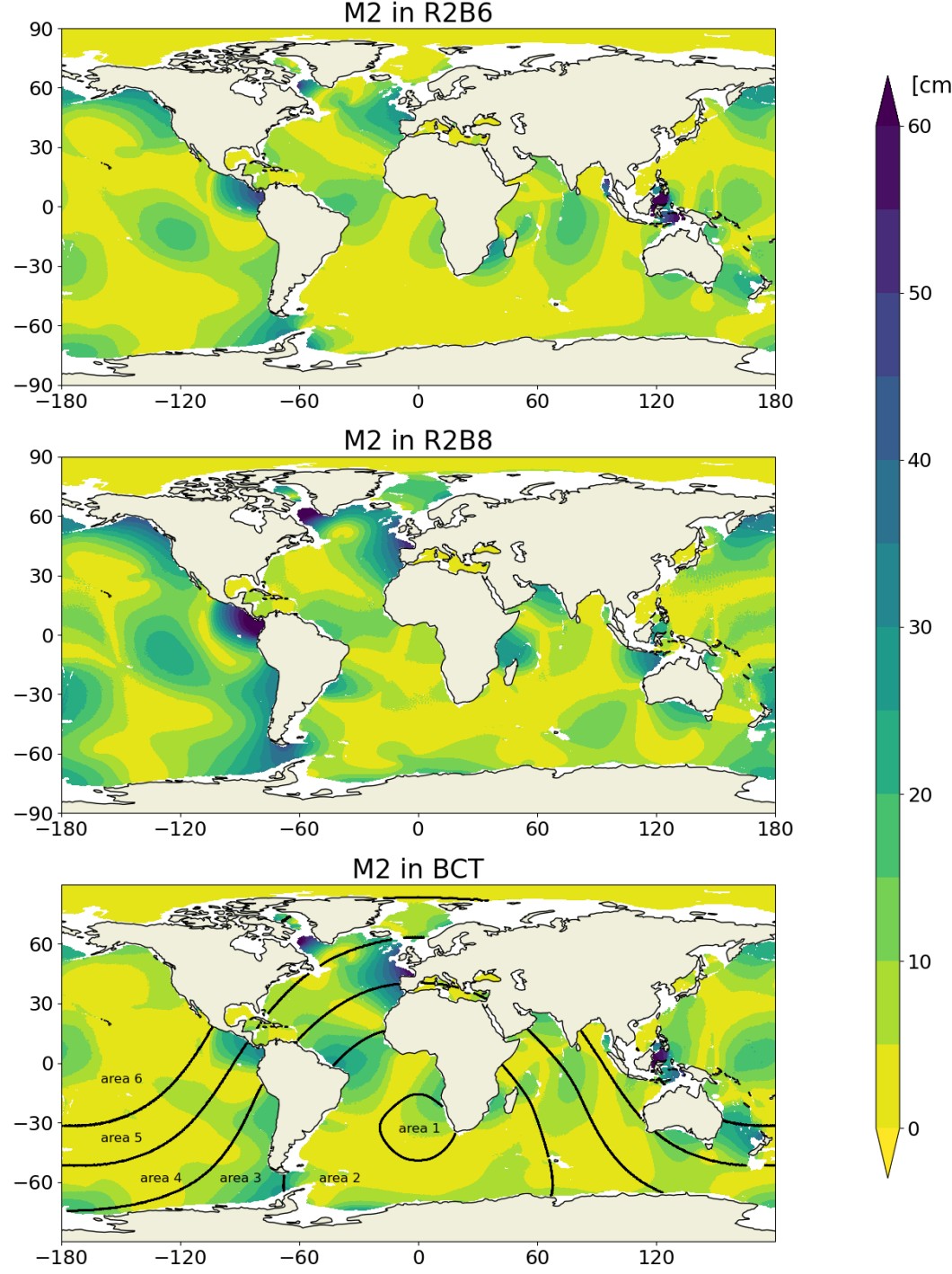

**Figure 8.** Same as Fig.4 but for the difference $d_{k,PH}$ resulting from errors in phase only.

| | $k = M_2$ | $S_2$ | $N_2$ | $K_2$ | $K_1$ | $O_1$ | $P_1$ | $Q_1$ | $\sum_{k=1}^{8}$ |
|---|---|---|---|---|---|---|---|---|---|
| $S_k$ (TPXO9) | 24.49 | 9.95 | 5.16 | 2.79 | 9.75 | 7.30 | 3.08 | 1.52 | |
| $S_k$ (ICON-O) | 31.67 | 12.96 | 5.55 | 3.47 | 10.84 | 9.25 | 3.59 | 1.58 | |
| $D_k \mid E$ | 14.25 | 7.90 | 2.30 | 2.30 | 4.60 | 5.19 | 1.49 | 0.86 | 63.38 |
| $D_{k,AM} \mid E_{AM}$ | 10.63 | 4.71 | 1.50 | 1.19 | 3.26 | 3.89 | 1.08 | 0.48 | 81.40 |
| $D_{k,PH} \mid E_{PH}$ | 9.49 | 6.34 | 1.74 | 1.96 | 3.25 | 3.43 | 1.02 | 0.72 | 81.97 |

**Table 2.** Values of signals, $D_k$, $D_{k,AM}$ and $D_{k,PH}$ (first eight columns of numbers) in cm, and the skills $E$, $E_{AM}$ and $E_{PH}$ (last column of numbers), as derived from the ICON-O in the standard R2B6 configuration. All values are averaged over grid points with water deeper than 1000 meters.

## 4 Effects of the horizontal resolution, spatial inhomogeneity and vertical coordinates

When increasing the horizontal resolution from about 40 km (R2B6) to about 10 km (R2B8), the error $d_k$ is not reduced for all
245 tidal constituents at all grid points. For the $M_2$ tide, the middle panel in Fig.6 shows that in many places the error seems to be enhanced, rather than reduced. When decomposing $d_k^2$ into $d_{k,AM}^2$ and $d_{k,PH}^2$, the middle panels of Fig.7 and Fig.8 show that increasing horizontal resolution leads to a general reduction of the error in amplitude, but worsens the error in phase, especially in the Pacific.

That increasing horizontal resolution does not improve the accuracy of the simulated $M_2$ tide in deep oceans is further
quantified in Fig.9, which compares for the $M_2$ tide the errors (top) and the skills (bottom) averaged over grid points shallower (right) with those averaged over grid points deeper than 1000 meters (left). Consider first the situation in the deep ocean. We see that the errors in simulating the amplitude of the $M_2$ tide ($D_{M_2,AM}$, top left panel) is reduced in R2B8 (red) relative to that in R2B6 (blue) as already suggested by Fig.7. This slight improvement is however outweighed by a larger phase error $D_{M_2,PH}$, making $D_{M2}$ to be larger in the R2B8 run than in the R2B6 run. When considering all eight tidal constituents (bottom panel),
the overall skill $E$ is lower in R2B8 (red, bottom left panel) than in R2B6 (blue). This somewhat lower skill in R2B8 is due to the lower skill in simulating the phase as measured by $E_{PH}$, while the skill in simulating the amplitude as measured by $E_{AM}$ is slightly enhanced.

A different picture is found for tides in shallow seas (right panel). For the $M_2$ tide, all three error measures $D_{M_2}$, $D_{M2,AM}$ and $D_{M2,PH}$ are smaller in R2B8 (red bars in the top right panel) than in R2B6 (blue bars in the top right panel). When
considering all constituents, all three skill scores $E$, $E_{AM}$ and $E_{PH}$ are higher in R2B8 (red bars in the bottom right panel) than in R2B6 (blue bars in the bottom right panel). We hence conclude that the major effect of increasing horizontal resolution is to improve the quality of the simulated tides in shallow seas, without having significant impact on the tides in the deep ocean.

Using a spatially inhomogeneous telescoping grid does not seriously deteriorating the quality of the simulated tides. The spatial distributions of $d_{M_2}$, $d_{M2,AM}$ and $d_{M2,PH}$ in BCT (bottom panels in Fig.6 - Fig.8) are comparable to those in R2B6
(top panels). Fig.9 shows further that in the deep oceans, the error related to the $M_2$ tide, $D_{M_2}$, is slightly reduced in BCT (orange) and the skill $E$ is slightly enhanced relative to the respective values in R2B6 (blue). In the shallow seas, the quality of

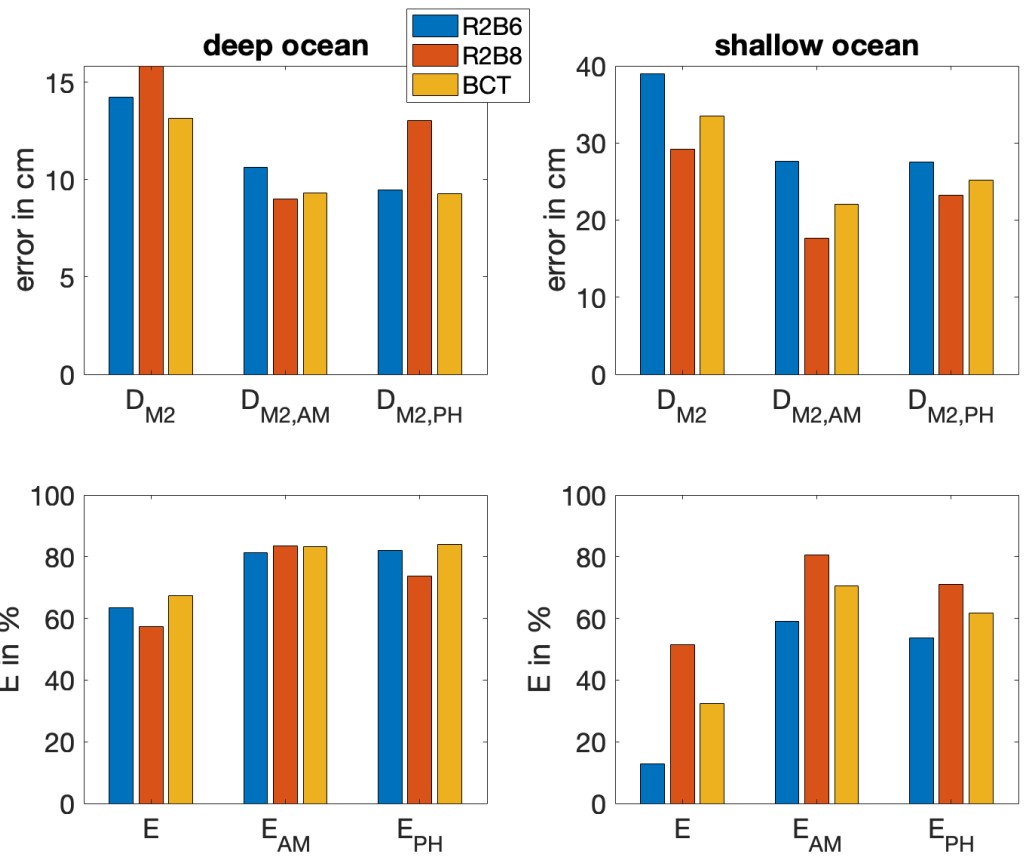

**Figure 9.** Effects of increasing horizontal resolution as quantified by the globally averaged errors $D_{M_2}$, $D_{M_2,AM}$, and $D_{M_2,PH}$ (top) for the $M_2$ tide, and by the skill scores $E$, $E_{AM}$, and $E_{PH}$ averaged over 8 constituents (bottom) simulated by ICON-O with a resolution of about 40 km (blue) and 10km (red) and with the BCT grid (orange). The bars in the left panel are obtained by averaging over grid points deeper than 1000 m, whereas the bars in the right panel are obtained by averaging over grid points shallower than 1000 m.

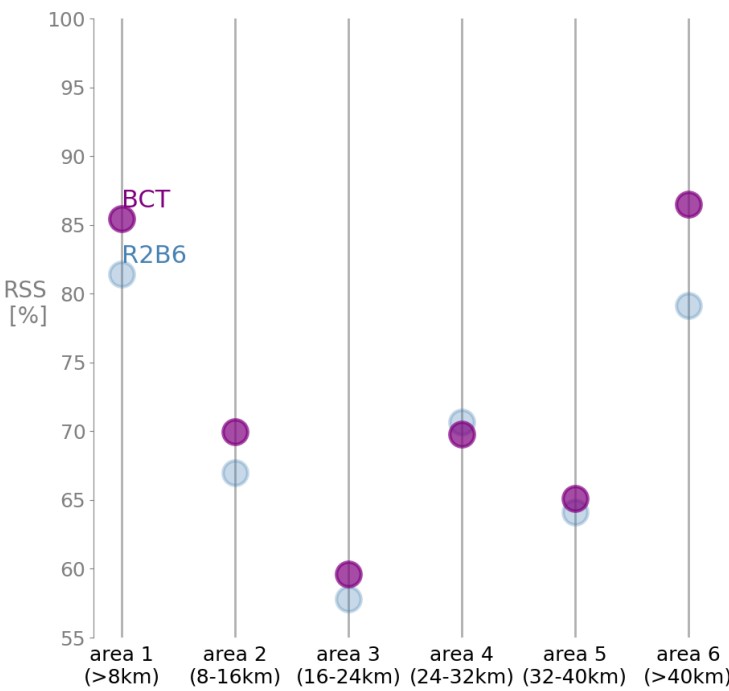

**Figure 10.** The overall skill score $E$ averaged over eight constituents in six different areas, with Area 1 containing grid boxes having the smallest sizes of less than 8 km and Area 6 containing grid boxes having the largest sizes of larger than 40 km.

the tides in BCT is also improved rather than deteriorated relative to R2B6, although to a less extent than in R2B8. In terms of skills, the values of $E$, $E_{AM}$ and $E_{PH}$ are higher than those in R2B6, but lower than those in R2B8. The result indicates that a spatially inhomogeneous telescope grid does not strongly deteriorate the quality of the simulated tides.

We further examine the dependence of the skill in BCT on the inhomogeneous grid size by calculating $E$ in BCT averaged over grid points in 6 selected areas, classified according to the grid sizes in the areas. The grid size increases from smaller than 8 km in Area 1 to grid size larger than 40 km in Area 6. The areas are indicated by the black lines in the bottom panel of Fig.6. We calculate the skill $E$ for each area and compare the result with the skill $E$ obtained from R2B6 in the respective area (Fig.10). For Area 1-3, which have grid sizes clearly smaller than the R2B6 grid size, the BCT tides are more accurate than the

R2B6 tides. The picture is less clear in Area 4 and 5, where BCT has grid size close to the R2B6 resolution. For some unknown reason, the BCT skill is higher than the R2B6 skill in Area 6, where the BCT grid sizes are larger than the R2B6 resolution. Fig.10 provides some additional confidence that the BCT grid does not seriously deteriorate the quality of the simulated tides.

We consider now the impact arising from switching from z-coordinate to z*-coordinate in the R2B6 configuration. We find hardly any change when switching from z-coordinate with a first-layer thickness of 14 meters to z*-coordinate with a first-layer thickness of 2 meters. The overall skill as measured by $E$ is 63.13%, almost identical 63.38% obtained using the z-coordinate.

## 5 Effect of topographic wave drag and SAL

Previous studies have shown that improving the accuracy of tides simulated by a barotropic tidal model requires the consideration of both the self-attraction and loading (SAL) effect (Hendershott (1972)) and the energy loss from barotropic tides to internal tides over rough topography (Jayne and St.Laurent (2001); Arbic et al. (2004)). The latter is often represented by a topographic wave drag (TWD). It is however not completely clear whether and how the parameterizations of SAL and TWD used in a barotropic tidal model that simulates *only* tides should be modified to get an accurate simulation of barotropic tides in an OGCM that simulates both tides and non-tidal motions. With respect to the TWD, a GCM with a horizontal resolution of about 10 km can simulate the conversion of barotropic tides to low-mode internal tides (Arbic et al. (2012); Li et al. (2015)). With such a GCM, part of the energy loss, namely the part that is transformed to the energy of the low-mode internal tides, is represented by the model and does not need to be parameterized separately. This seems to suggest that we still need to parameterize the energy loss to high-mode internal tides. When using the 102 pelagic tidal gauges as reference, the skill $E$ of the simulated tides is 92.8% for a MPIOM simulation that does not include a TWD (Mueller et al. (2012)) and 93% for a HIM (Hallberg Isopycnal Model) simulation that includes a TWD (Arbic et al. (2004)). These results do not suggest a decisive role of the inclusion of the energy loss to high-mode internal tides in improving the accuracy of the simulated tides.

In subsection 5.1 and 5.2 we describe the two parameterizations implemented in ICON-O: The topographic wave drag (TWD) that paramterizes the energy loss over rough topography from barotropic tides to internal tides and the parameterization of SAL in form of a simple scalar approximation by Ray (1998). The impacts of these parameterizations on the quality of the simulated tides are quantified in subsection 5.3.

### 5.1 Topographic wave drag

The energy conversion from barotropic tides to internal tides, which causes energy loss of barotropic tides, can be parameterized as an additional bottom drag. In addition to the standard quadratic bottom drag, $c_B|\mathbf{u}_B|\mathbf{u}_B$, with $c_B$ being a bottom drag coefficient and $\mathbf{u}_B$ the bottom velocity, we add the linear topographic wave drag:

$$\mathcal{D}_{TWD} = \frac{1}{2}\kappa h^2 N_B \mathbf{u}_B \tag{9}$$

The form of this drag is motivated by the scaling relation of the energy conversion from barotropic tides to internal tides (Jayne and St.Laurent (2001)). Here, $h^2$ represents the bottom roughness which we compute as the topography variance within a 100 km radius around each grid point, and $N_B$ is the bottom stratification.

Zarroug et al. (2010) argue that using the stratification averaged over some vertical distance above the ocean floor instead of using the bottom stratification $N_B$ provides a better estimate of the internal tide generation and therefore also a more realistic

topographic wave drag. In this scenario, the respective vertical distance should be linked to the vertical scale of the locally dominant internal tide mode. However, in this first attempt to modeling tides in ICON-O, we use the most widely adopted version based on the bottom stratification only.

$\kappa$ represents a typical horizontal wavelength of the generated internal tides and is in reality dependent on the dominant topographic length scale. Within the framework of the TWD parameterization, $\kappa$ is used as a geographically homogeneous tuning parameter that allows to minimize the error of the modeled barotropic tide. It turns out that in ICON-O, the best result can be obtained when choosing $\kappa = 50$ km. This is in the range of what we expect as typical wavelength for internal tides and also what was used previously (Jayne and St.Laurent (2001); Exarchou et al. (2012)).

$\mathbf{u}_B$ in Eq.(9) stands for the bottom flow. Ideally, $\mathcal{D}_{TWD}$ should remove only energy from the *tidal* flow and not from the non-tidal bottom flow, implying that $\mathbf{u}_B$ should contain only the tidal bottom flow. For this purpose, Arbic et al. (2010) apply a 24-hour running-mean filter to the bottom flow at each timestep that approximately separates the tidal from the non-tidal flow. They then apply the drag only to the tidal flow.

The importance of such a data-intensive online filtering can be estimated by comparing the energy that is removed by the TWD when using as $\mathbf{u}_B$ firstly the full bottom flow (including tidal and non-tidal flow) and secondly only tidal flow (obtained from the reconstruction based on the harmonic analysis). According to this analysis which we performed for the R2B6 simulation, the non-tidal bottom flow accounts for only about 5% of the energy removed by the TWD, indicating that the tidal flow is by far the dominant flow in the abyssal oceans. We conclude that using the full bottom flow as $\mathbf{u}_B$ in TWD parameterization will not lead to an unreasonably large error. This allows us to not apply the online time filtering and thereby keeping the parameterization in its simplest form.

## 5.2 Self-attraction and Loading (SAL)

The parameterization of SAL is supposed to cover the effects of deformation of the ocean seafloor due to the weight of the water column, the associated mass redistribution and the according changes in the gravitational field, and the gravitational attraction of the water body on itself. Generally, a more complete description of SAL should utilize a decomposition of mass anomalies into their spherical harmonic constituents, as it is done in Gordeev et al. (1977), Brus et al. (2012), Barton et al. (2022) and Shihora et al. (2022). As a first attempt however, we use here the scalar approximation of the SAL effect (Ray (1998)), which is less accurate as the consideration based on spherical harmonics.

Generally, SAL can arise from all barotropic motions, including non-tidal ones, especially those at high-frequencies. For low frequencies motions, it is quite save to assume that there is a quasi-instantaneous adaptation of the sea water to the time-variable external gravity field of the Earth (Shihora et al. (2022)). This timescale dependence of the effect of SAL can lead to complications when implementing SAL in an OGCM that simulates both non-tidal and tidal flows. Being expressed as an additional body force, a SAL parameterization impacts simulated motions on all time scales. One cannot expect the same parameterization, that works well for a barotropic tidal model, to work also well for an OGCM. In the study by Shihora et al. (2022), implementing a SAL parameterization into an OGCM does not improve the accuracy of the simulated tides to the same

degree as it does when implementing the same SAL parameterization into a barotropic tidal model, suggesting the complexity of the problem and the value of first concentrating on the simple scalar approach.

Following the scalar approach (Ray (1998)), we add to the astronomical tide potential an additional SAL potential that is proportional to tide-induced sea surface height (SSH) $\eta_{tidal}$,

$$\eta_{SAL} = \beta \eta_{tidal} \tag{10}$$

with $\beta = 0.06$. While the model SSH $\eta$ is equivalent to $\eta_{tidal}$ in pure tidal models (e.g. Jayne and St.Laurent (2001)), this is no longer true in a 3-dimensional OGCM that has a heterogeneous stratification and non-tidal flow components (Arbic et al. (2010)). To keep things simple, we consider an approximation of the tide-induced SSH $\eta_{tidal}$ by correcting the full model SSH $\eta$ for the two largest non-tidal SSH contributions:

$$\eta_{tidal} \approx \eta - \eta_{tm} - \eta_{steric} \tag{11}$$

where $\eta_{tm}$ is the contribution due to the large-scale, time-mean circulation and $\eta_{steric}$ the so-called steric contribution. We take $\eta_{tm}$ as the time-mean model SSH of the last two years of the spinup runs. Removing $\eta_{tm}$ is consistent with the expectation that the SAL effect does not arise from low-frequency motions. The steric contribution is computed as

$$\eta_{steric} = \int_0^H \frac{<\rho> - \rho}{\rho_0} \, dz, \tag{12}$$

where $H$ is the ocean depth, $<\rho>$ is the time-mean of density $\rho$ and $\rho_0 = 1025$ kg m$^{-3}$ is a reference density. Correcting for the steric SSH contribution efficiently filters out the SSH imprint of time-varying baroclinic motions such as those related to mesoscale eddies. This way, we can make sure that our additional SAL potential does not introduce too much spurious effects due to the time-mean circulation and the non-tidal baroclinic motion in ICON-O.

## 5.3 Results

Further tidal simulations (indicated by the non-bold letters in Tab.1) are carried out, with either one of the two or with both parameterizations switched on. We consider the three configurations described in Section 3, namely the standard R2B6 configuration, the higher-resolution configuration with the R2B8 grid, and the configuration with the inhomogenous BCT grid. These simulations are then evaluated against the TPXO9 by calculating the skill defined in Eq.(7), both $E$ related to the total errors, and $E_{AM}$ and $E_{PH}$ related to the amplitude and phase errors. The results are shown in Fig.11.

We found that the two parameterizations (Fig.11) have different effects on tides in deep and shallow oceans. Consider first tides in deep oceans (left column of Fig.11). With respect to the overall skill $E$ (top left panel), the largest difference within each group of bars is found for the R2B8 configuration (middle), suggesting that the two parameterizations have the largest impact on the simulated tides in R2B8 than in R2B6 and in BCT configuration. For R2B8, E increases from barely 60% (blue) without SAL and TWD to about 71% when including SAL (red) and to about 74% when including both parameterizations (magenta). For R2B6, E increases from 63% (blue) without SAL and TWD to 69.5% (orange) when employing TWD, while

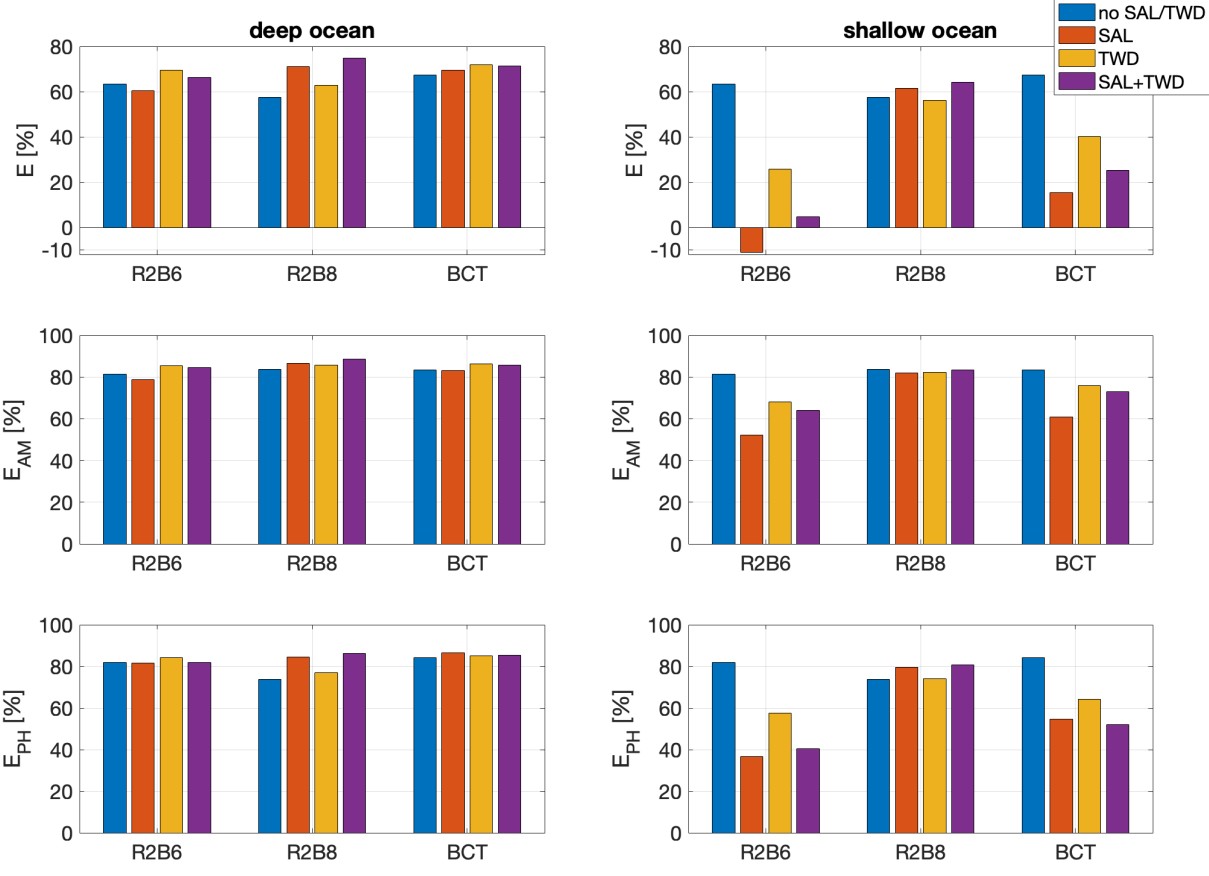

**Figure 11.** Effects of parameterizations SAL and TWD as quantified by the skill scores $E$ (top), $E_{AM}$ (middle), and $E_{PH}$ (bottom) for ICON-O in R2B6, R2B8 and BCT configurations. Blue bars indicate the case without employing SAL and TWD, red bars the case with SAL only, orange bars the case with TWD only, and magenta bars the case with both SAL and TWD. The values are obtained by averaging over eight constituents and grid points deeper (left) and shallower (right) than 1000 m.

employing SAL (red) slighly degrades the simulated tides. Both SAL and TWD together (magenta) still give an improvement for R2B6. For BCT, the skill is only slightly enhanced when employing both parameterizations (individually and together).

The improvement achieved with the R2B8 configuration arises mainly from including the SAL parameterization. As suggested by $E_{AM}$ and $E_{PH}$ for R2B8, the SAL-induced improvement is related to a reduction in both amplitude and phase errors, leading to larger values of $E_{PH}$ and $E_{AM}$, with the increase in $E_{PH}$ being more prominent than that in $E_{AM}$. The result is consistent with the previous consideration that the SAL can affect both the amplitudes and the phases (Ray (1998)).

With the R2B6 configuration, the improvement achieved arises mainly from including the TWD parameterization, which damps the simulated tides. This alleviates the overestimation of the tidal signals, leading to a higher skill $E_{AM}$ with TWD than without TWD. When using the BCT configuration, the SAL parameterization slightly reduces the phase error and with that slightly increases the skill $E_{PH}$. The TWD parameterization slightly reduces the amplitude error and with that slightly increases the skill $E_{AM}$.

The situation is quite different in shallow oceans (right column of Fig.11). There, the SAL parameterization does not work properly in a low relation configuration. We find a strong reduction of skill $E_{AM}$ and skill $E_{PH}$ in R2B6 and in BCT configuration. Recall that relative to R2B8, BCT has a coarser resolution over a major part of the world ocean. The total skill $E$ is even negative in R2B6. The TWD parameterization, when implemented in low-resolution configuration, also degrades the skills, albeit the degradation is less server compared to that induced by SAL. Such strong degradation is not found in the R2B8 configuration. The SAL parameterization in R2B8 is able to reduce the phase errors, thereby enhancing $E_{PH}$, not only in deep but also in shallow oceans.

## 6 Discussions

This paper suggests that for ICON-O with 128 vertical levels, the quality of open-ocean tides cannot be satisfactorily improved by including parameterizations of SAL and TWD. Generally, when implementing parameterizations of SAL and TWD developed for a single-layer model (Egbert et al. (2004), Gordeev et al. (1977), Pal et al. (2023)) in a more realistic baroclinic OGCM, some additional procedures are required. For TWD, which is a drag acting on tidal velocity only, we need a procedure that separates tidal velocity from the full velocity. For SAL, we need a procedure that separates sea level height variations arising from high-frequency barotropic motions from those arising from the large-scale time-mean circulation and low-frequency baroclinic features. Arbic et al. (2010) addressed this problem. Shihora et al. (2022) showed that implementing a parameterization of SAL in a GCM is less successful compared to the case for a barotropic tidal model. Here we try out the simplest approach. For TWD, we compare the energy removed by TWD equipped with full bottom velocity with the energy removed by TWD equipped with tidal velocity only, and conclude that the error arising by using full (instead of tidal) velocity is negligible. For SAL, we use only the seal level height obtained by removing the time-mean sea level height and a steric contribution computed as a vertical integral of normalized density anomaly. Further investigation is needed to figure out whether the quality of tides simulated by ICON-O can be further improved by optimizing the separation procedures.

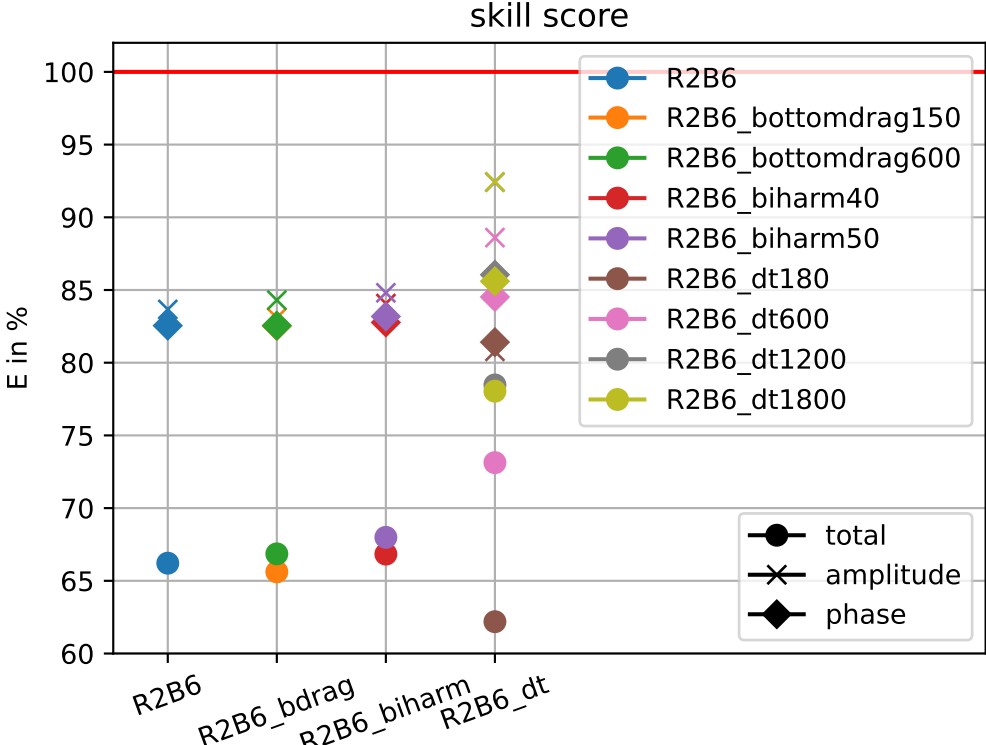

**Figure 12.** Sensitivity of skill score $E$ to bottom drag, biharmonic viscosity and time step obtained from ICON-O in R2B6 configuration. Shown are 9 simulations performed with R2B6 ICON-O on the new high performance computer Levante at DKRZ, which has replaced Mistral recently. One corresponds to the R2B6 run described before (blue); two with the strength of bottom drag being doubled (green) and halved (orange); two with biharmonic viscosity being increased by 14% (red) and by 40% (purple); four with time step being reduced from 300 seconds to 180 second (brown) and increased from 300 seconds to 600 (pink), to 1200 seconds (grey), and to 1800 seconds (yellow-green).

Apart from the SAL and TWD parameterizations, the quality of tides can also be affected by other parameters in the model. We hence performed a suite of additional experiments to address the sensitivity of the quality of the simulated tides to some of these parameters. Given that the tidal signals in our standard R2B6 configuration are too strong (first two rows in Tab.2), we paid special attention to the following parameters that have the potential to affect the strength of barotropic motions: the strength of the bottom drag, the biharmonic viscosity coefficient, and the time step. To this end, we point out that the time stepping scheme in ICON-O (Korn (2017)) has a damping effect, which strengthens with increasing time step for fixed values of parameters (i.e. for fixed values of $\beta$ and $\gamma$ in Eq.(33) and Eq.(34) in Korn (2017)). To exclude the time step sensitivity, we used in all experiments summarized in Tab.1 the same values of $\beta$ and $\gamma$ and the same time step of 300 seconds which is required for a stable R2B8 simulation.

Fig.12 shows the sensitivity of the skill $E$ to bottom drag, biharmonic viscosity, and time step obtained from this new suite of experiments. We found that doubling and halving the bottom drag coefficient have only a small effect on the quality of the simulated tides. The same is true for increasing the biharmonic viscosity. In contrast, we find larger changes of skill $E$ when we alter the model time step. With a time step of 1800 seconds, which would be normally used in R2B6 configuration, the overall skill $E$ increases from about 65% (blue bullet) to almost 80% (yellow-green bullet). When considering only the errors in amplitudes, the skill $E_{AM}$ is found to be well above 90% (yellow-green cross). Similar skill are obtained with a time step of 300 second, when the values of $\beta$ and $\gamma$ are properly adjusted (not shown).

## 7 Summary

This paper assesses the ability of a newly developed OGCM - ICON-O in realistically simulating open-ocean tides. In the standard R2B6 configuration with a horizontal resolution of 40 km, ICON-O is able to simulate the main features of the tides, as found in the geographical distributions of amplitude, phase and amphidromic points obtained from TPXO9. When keeping the time step and the parameters in the time stepping scheme the same as in R2B8 with finer resolution, the overall skill $E$ (averaged over eight constituents) is about 63% against TPXO9 and about 77% against pelagic station data. The skill $E$ increases to about 80% when adjusting the time step or the parameters in the time stepping scheme. Using the R2B8 configuration, where the horizontal resolution is increased to 10 km, leads to an overall improvement of tides in coastal regions, but not in the deep oceans. Using a spatially highly inhomogenous grid does not deteriorate the quality of the simulated tides. The transformation from ICON-O based on z-coordinate to ICON-O based on z*-coordinate is successful. The latter allows simulating tides with the same accuracy without employing a thick surface layer.

The effect of SAL and TWD parameterizations on the simulated tides depends on the configurations considered. The SAL parameterization, when implemented in the R2B8 resolution, noticeably reduces the phase error and enhances the accuracy of the simulated tides both in the deep and in the shallow oceans. The situation is different when implementing SAL in a lower resolution configuration. In R2B6, no improvement can be achieved for tides in deep oceans, while strong degradation is found for tides in shallow oceans. In BCT, there is a slight improvement of the skill in deep oceans, but a clear reduction of skill in shallow oceans. The TWD parameterization damps the overestimated tidal amplitudes, whereby reducing the amplitude error and enhancing the accuracy of the simulated deep-ocean tides in all three configurations (R2B6, R2B8 and BCT). In shallow oceans, TWD has little effect in R2B8 configuration, but degrade the tides in both R2B6 and BCT configuration. The result suggests that when interested only in deep-ocean tides, the TWD parameterization should be included independent of the model configuration. It is not meaningful to study shallow-ocean tides using a low-resolution model with the SAL parameterization included.

The error- and skill-analyses were carried out for grid points deeper than the threshold $d_*$ of 1000 m. This value of $d_*$ is chosen quite arbitrarily. Varying $d_*$ does not change the overall picture. In particular, when $d_*$ is slightly reduced, the strength of the observed tidal signal increases and the root sum square of errors also increases, leaving the skill $E$ as defined in Eq.(7) largely unchanged.

The purpose of assessing the ICON-O's ability in realistically simulating the open-ocean tides is to ensure that ICON-O in km-scale resolution is able to realistically simulate internal tides, which are much more difficult to evaluate than the barotropic tides. The work reported here presents only the first efforts. Even though ICON-O has somewhat lower skill in simulating barotropic tides than MPIOM and HYCOM, we believe that the barotropic tides simulated by ICON-O are accurate enough for studying some statistics of internal tides, such as the energy fluxes related to internal tides. For such studies, the phase errors are not crucial. When taking out the phase errors and properly choosing the parameters in the time stepping scheme, the skill, as measured by $E_{AM}$, can be well above 90%. Although encouraging, there is plenty room for further improvements.

*Code availability.*

Simulations were done with ICON-O version icon-2.6.6. This source code is available on Edmond – the Open Research Data Repository of the Max Planck Society under: https://doi.org/10.17617/3.VQB9N8 (von Storch et al. 2023). The ICON model is available to individuals under the licenses provided under the above link. By downloading the ICON source code, the user accepts the license agreement.

*Data availability.*

The tidal simulations presented in this paper are published at the long-term archive (DOKU) at DKRZ: http://hdl.handle.net/21.14106/a2c5432b7cb7f3a16d126dea36267d01d2abb3d6 (Hertwig et al. 2021). These simulations were started from states at the end of the respective spinup simulations published also at the long-term archive (DOKU) at DKRZ: http://hdl.handle.net/21.14106/46e641035ea657a2f90f7ebe6501d327643d1087 (Hertwig et al. 2021). The results of the harmonic analysis (amplitudes and phases of 8 tidal constituents) are published at the World Data Center for Climate (WDCC) at DKRZ: https://doi.org/10.26050/WDCC/Tides_ICON-O (Hertwig et al.2022) The scripts used for the analysis are published on Zenodo: https://doi.org/10.5281/zenodo.8085145 (Hertwig and von Storch, 2023), and the TPXO9 data used for ICON-O tides evaluation on Zenodo: https://doi.org/10.5281/zenodo.8074917 (Hertwig and von Storch, 2023).

*Author contributions.*

J.-S. von Storch designed the experiments, performed some of the evaluation analysis, wrote the manuscript

E. Hertwig carried out the experiments, performed all harmonic analyses and most of the evaluation analyses, curated the data, edited the manuscript

V. Lüschow implemented the SAL and the TWD parameterizations into ICON-O

N. Brüggemann edited the manuscript

N. Brüggemann and H. Haak assisted all, performed some experiments, did some of the evaluation analysis, and curated the model code

P. Korn assisted the experiments and provided suggestions to the manuscript

V. Singh implemented z*-coordinate

*Competing interests.*

The authors declare that they have no conflict of interest

*Acknowledgements.* This study is a contribution to project W2 (Scattering and Refraction of Low-Mode Internal Tides by InteractionWith
Mesoscale Eddies) of the Collaborative Research Centre TRR 181 "Energy Transfer in Atmosphere and Ocean" funded by the Deutsche
Forschungsgemeinschaft (DFG, German Research Foundation)—Project number 274762653.

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
