# Peer review of "Open-Ocean Tides Simulated by ICON-O, Version icon-2.6.6"

_EGUsphere, 2023_

## Referee Comment (RC1)

General Comments:

The paper presents results of barotropic tidal dynamics simulated using a newly developed ocean general circulation model ICON. In particular, the paper discusses the effect of various grid resolutions, and the inclusion of two specific processes, namely tidal bottom drag (TBD) and Self Attraction and Loading (SAL), on the barotropic tidal dynamics. The paper is clearly written, and has scientific relevance to the broader ocean modeling community. However, most of the dynamics discussed in this paper have been looked at in some detail in recent publications. For example, use of tides in an OGCM has been investigated in MPAS-Ocean (the ocean component of the Energy Exascale Earth System Model) (e.g., "Barotropic tides in MPAS-Ocean (E3SM V2): impact of ice shelf cavities", Nairita Pal, Kristin N. Barton, Mark R. Petersen, Steven R. Brus, Darren Engwirda, Brian K. Arbic, Andrew F. Roberts, Joannes J. Westerink, Damrongsak Wirasaet; Geoscientific Model Development 16 (4), 1297–1314, 2023). The model in that paper has a topographic wave drag (what the authors call TBD in this manuscript) and self attraction and loading (SAL), very similar to what is reported in the current manuscript. Therefore, it might be better to explore the ICON model in some more detail and explain how tides in ICON-O are unique/different from several other existing OGCMs. The authors should include some more science explanations behind the model results (e.g., the reasons behind why the tidal errors occur in the specific locations). Once these points are addressed, I will be happy to read the revised manuscript. Some specific issues are discussed below.

Specific comments:

1.  Please include references to published works exactly aligned with the current manuscript. Especially, in Line number 45 the authors mention "T*o our knowledge, modelling tides using global OGCMs (for the purpose of studying internal tides) have only been carried out in HYCOM simulations using the HYbrid Coordinate Ocean Model (Arbic et al. (2010); Arbic et al. (2012)) and in STORMTIDE / STORMTIDE2 using the Max-Planck Ocean Model (MPIOM) (Mueller et al. (2012); Li and von Storch (2020))"* . This is only partially true, since recently (2022 and 2023) tides have been studied in the MPAS-Ocean global model. The specific references are listed here:
    a)  "Barotropic tides in MPAS-Ocean (E3SM V2): impact of ice shelf cavities", Nairita Pal, Kristin N. Barton, Mark R. Petersen, Steven R. Brus, Darren Engwirda, Brian K. Arbic, Andrew F. Roberts, Joannes J. Westerink, Damrongsak Wirasaet; Geoscientific Model Development 16 (4), 1297–1314, 2023
    b)  "Scalable self attraction and loading calculations for unstructured ocean tide models"; Steven R Brus, Kristin N Barton, Nairita Pal, Andrew F Roberts, Darren Engwirda, Mark R Petersen, Brian K Arbic, Damrongsak Wirasaet, Joannes J Westerink, Michael Schindelegger; Ocean Modelling, 102160, 2023
    c)  "Global barotropic tide modeling using inline self-attraction and loading in MPAS-Ocean";Kristin N Barton, Nairita Pal, Steven R Brus, Mark R Petersen, Brian K Arbic, Darren Engwirda, Andrew F Roberts, Joannes J Westerink, Damrongsak Wirasaet, Michael Schindelegger; Journal of Advances in Modeling

Earth Systems, e2022MS003207; 2022.

The same comment holds for the paragraph after line 275. Please mention notable works.

2. As I understand, in Fig. 9, there is no SAL and TBD. In Fig.11 however, there is SAL and TBD. It might be good to see the corresponding figures for Fig. 9 (a) and (b) when SAL and TBD are included. Fig.11 can be removed altogether, and instead, in Fig.9(a), (b), (c), (d) a comparison of SAL/TBD and no SAL/TBD can be done for deep and shallow ocean cases.

3. It might be good to see the corresponding figures of Figs. 6, 7, 8 with SAL/TBD effects included as well. It might help us know if SAL/TBD improves M2 amplitude / phase errors.

4. The use of the term "tidal bottom drag" is slightly confusing, since bottom drag generally refers to a friction with the bottom boundary layer (e.g., as the authors mention just before line 295, a quadratic drag). Here the tidal bottom drag possibly refers to the internal wave drag over rough topography as mentioned in Jayne and St. Laurent (2001), and also in reference 1 mentioned above. An explanation is needed of why the authors use the term "tidal bottom drag" instead of the standard internal or topographic wave drag.

5. In the paragraph of line 305, the authors mention that they choose \kappa=50 km. I was wondering if the authors have explored experiments with other values of \kappa. More specifically, do the M2 phase errors and amplitude errors reduce when other values of \kappa are used?

6. Figs. 6, 7, 8 show that the errors in the Drake passage are considerably reduced when using BCT grid. The resolution there seems to be between 24 to 32km. However, the resolution of the R2B8 grid is 10km uniform globally. I was wondering why the errors improve in the Drake passage (for the BCT grid) even though the resolution is coarser than the R2B8 grid.

7. Why are the errors so high in the Southern Ocean? Please elaborate.

8. In Fig. 9(a) why are the M2 phase errors for R2B8 considerably higher than R2B6 in the deep ocean? Does it point towards any processes incorrectly captured?

Typographical errors

1. Line 138 "Hy- drographic" → "Hydrographic"

Overall, the paper is well written. It might be useful to include the science behind the observed results for better impact.

---

## Author Response (AR1)

**Reply to RC1**

General Comments:

The paper presents results of barotropic tidal dynamics simulated using a newly developed ocean general circulation model ICON. In particular, the paper discusses the effect of various grid resolutions, and the inclusion of two specific processes, namely tidal bottom drag (TBD) and Self Attraction and Loading (SAL), on the barotropic tidal dynamics. The paper is clearly written, and has scientific relevance to the broader ocean modeling community. However, most of the dynamics discussed in this paper have been looked at in some detail in recent publications. For example, use of tides in an OGCM has been investigated in MPAS-Ocean (the ocean component of the Energy Exascale Earth System Model) (e.g., "Barotropic tides in MPAS-Ocean (E3SM V2): impact of ice shelf cavities", Nairita Pal, Kristin N. Barton, Mark R. Petersen, Steven R. Brus, Darren Engwirda, Brian K. Arbic, Andrew F. Roberts, Joannes J. Westerink, Damrongsak Wirasaet; Geoscientific Model Development 16 (4), 1297–1314, 2023). The model in that paper has a topographic wave drag (what the authors call TBD in this manuscript) and self attraction and loading (SAL), very similar to what is reported in the current manuscript. Therefore, it might be better to explore the ICON model in some more detail and explain how tides in ICON-O are unique/different from several other existing OGCMs. The authors should include some more science explanations behind the model results (e.g., the reasons behind why the tidal errors occur in the specific locations). Once these points are addressed, I will be happy to read the revised manuscript. Some specific issues are discussed below.

We thank the reviewer for his/her comments, in particular for pointing out the recent work on tides in MPAS-Ocean. We notice that different from the work with MPAS-O, which focuses on barotropic tides by considering a single-layer version of MPAS-O, our ultimate goal is to develop an OGCM capable for investigating internal tides. In our work, evaluating the quality of barotropic tides in ICON-O is considered as a first necessary step toward achieving this goal, as it is much harder to evaluate internal tides compared with barotropic tides – the driving forcing for internal tides. On the other hand, internal tides can only be studied using a multi-layer baroclinic ocean model. We hence examine tides in ICON-O with 128 vertical levels. In our response to RC2, we point out that even though both SAL and topographic wave drag (TWD) are known for a long time, parameterizations of SAL and TWD are often only implemented and evaluated in a barotropic model that simulates only tidal motions. How these parameterizations should be modified when implemented in a multi-layer GCM that simulates both tidal and non-tidal motions is not completely clear.

In the new discussion section – Section 6, we now address this problem explicitly. With that we hope to raise the awareness in the community that implementing SAL and TWD into a multi-layer GCM may not be that straightforward and requires more

efforts to improve the quality of tides in a baroclinic GCM. To our knowledge, only the HYCOM group has systematically investigated this problem. Here, with the newly developed ICON-O, we report another effort inspired by Arbic et al. (2010) but considering also the effects of grid configurations (e.g. the telescope grid).

In the new Section 6, we discuss also a new set of experiments aimed to assess the sensitivity of the quality of the simulated tides to several other model parameters. We find strong impact of the time step scheme on the quality of the tides. In general, the time stepping scheme in ICON-O has a damping effect, whose strength depends on the time step and the two parameters in the time stepping scheme. We could reduce our too strong tidal amplitudes in our R2B6 configuration by increasing the time step or by adjusting the two parameters in the time stepping scheme, thereby significantly improve the skill in our R2B6 configuration. Apart from the time step and the parameters in the time stepping scheme, the other considered parameters do not show strong effect on the tides. We hope that this additional discussion helps interpreting our results and the sensitivity of the model.

Specific comments:

1. Please include references to published works exactly aligned with the current manuscript. Especially, in Line number 45 the authors mention "To *our knowledge, modelling tides using global OGCMs (for the purpose of studying internal tides) have only been carried out in HYCOM simulations using the HYbrid Coordinate Ocean Model (Arbic et al. (2010); Arbic et al. (2012)) and in STORMTIDE / STORMTIDE2 using the Max-Planck Ocean Model (MPIOM) (Mueller et al. (2012); Li and von Storch (2020))"*. This is only partially true, since recently (2022 and 2023) tides have been studied in the MPAS-Ocean global model. The specific references are listed here:

a) "Barotropic tides in MPAS-Ocean (E3SM V2): impact of ice shelf cavities", Nairita Pal, Kristin N. Barton, Mark R.        Petersen, Steven R. Brus, Darren Engwirda, Brian K. Arbic, Andrew F. Roberts, Joannes J. Westerink, Damrongsak Wirasaet; Geoscientific Model Development 16 (4), 1297–1314, 2023

b) "Scalable self attraction and loading calculations for unstructured ocean tide models"; Steven R Brus, Kristin N Barton, Nairita Pal, Andrew F Roberts, Darren Engwirda, Mark R Petersen, Brian K Arbic, Damrongsak Wirasaet, Joannes J Westerink, Michael Schindelegger; Ocean Modelling, 102160, 2023

c) "Global barotropic tide modeling using inline self-attraction and loading in MPAS-Ocean";Kristin N Barton, Nairita Pal, Steven R Brus, Mark R Petersen, Brian K Arbic, Darren Engwirda, Andrew F Roberts, Joannes J Westerink, Damrongsak Wirasaet, Michael Schindelegger; Journal of Advances in Modeling Earth Systems, e2022MS003207; 2022.

The same comment holds for the paragraph after line 275. Please mention notable works.

*We included the three papers (Brus et al. and Barton et al. in the first paragraph of section 5.2, Pal et al. and Barton et al. in the first paragraph of section 6). We also changed the sentence "To our knowledge, modelling tides using global OGCMs (for the purpose of studying internal tides) have only been carried out in HYCOM..." to "To our knowledge, modelling tides using multi-layer OGCMs (for the purpose of studying internal tides) have only been carried out in HYCOM...".*

2. As I understand, in Fig. 9, there is no SAL and TBD. In Fig.11 however, there is SAL and TBD. It might be good to see the corresponding figures for Fig. 9 (a) and (b) when SAL and TBD are included. Fig.11 can be removed altogether, and instead, in Fig.9(a), (b), (c), (d) a comparison of SAL/TBD and no SAL/TBD can be done for deep and shallow ocean cases.

*Thanks for the suggestion. We now included a consideration of SAL and TWD in both deep and shallow region as in Fig.9. However, we did so using the format of Fig.11, rather than that of Fig.9. This is because Fig.9 shows both the error D and the skill E; and D and E are related with a larger value of D being corresponding to a smaller value of E. To limit the number of figures, we choose to present the result in terms of the skill E only, as it is done in Fig.11. We see from the new Fig.11 that the parameterizations of SAL and TWD, especially that of SAL, do not improve the accuracy of tides in shallow regions. We discuss the situation in shallow oceans in the last paragraph of Section 5.*

3. It might be good to see the corresponding figures of Figs. 6, 7, 8 with SAL/TBD effects included as well. It might help us know if SAL/TBD improves M2 amplitude / phase errors.

*We included these plots at the end of this response letter. We see not only different effects of SAL/TWD in different regions, e.g. that in R2B6, SAL reduces the phase errors in the central and western Pacific and in the Drake Passage but enhances the phase errors in the Mozambique Channel and in the Weddell Sea (top panel in Fig.RC1-3). We see also some general effects already captured by Fig.11, e.g. that TWD reduces the amplitude errors in all three configurations (R2B6, R2B8 and BCT) (Fig.RC1-5). Since we are not very confident in explaining the different effects of SAL/TWD in different regions, and since we wish to keep the paper as concise as possible, these plots are not included in the paper.*

4. The use of the term "tidal bottom drag" is slightly confusing, since bottom drag generally refers to a friction with the bottom boundary layer (e.g., as the authors mention just before line 295, a quadratic drag). Here the tidal bottom drag possibly refers to the internal wave drag over rough topography as mentioned in Jayne and St. Laurent (2001), and also in reference 1 mentioned above. An explanation is

needed of why the authors use the term "tidal bottom drag" instead of the standard internal or topographic wave drag.

We now replaced the term TBD by TWD, both in the manuscript and in this response letter.

5. In the paragraph of line 305, the authors mention that they choose \kappa=50 km. I was wondering if the authors have explored experiments with other values of \kappa. More specifically, do the M2 phase errors and amplitude errors reduce when other values of \kappa are used?

This is a good suggestion. We included this as a direction for further investigations in the new Section 6.

6.Figs. 6, 7, 8 show that the errors in the Drake passage are considerably reduced when using BCT grid. The resolution there seems to be between 24 to 32km. However, the resolution of the R2B8 grid is 10km uniform globally. I was wondering why the errors improve in the Drake passage (for the BCT grid) even though the resolution is coarser than the R2B8 grid.

A good question. We think that resolution is just one factor. There may exist other factors that also affect the errors.

7. Why are the errors so high in the Southern Ocean? Please elaborate.

We add a possible explanation following Pal et al. in the third paragraph of Section 3.

8. In Fig. 9(a) why are the M2 phase errors for R2B8 considerably higher than R2B6 in the deep ocean? Does it point towards any processes incorrectly captured?

 The problem may be more complex due to resolved internal tides

Typographical errors

1.  Line 138 "Hy- drographic" → "Hydrographic"

Corrected

Overall, the paper is well written. It might be useful to include the science behind the observed results for better impact.

We made an attempt by including some discussions and a new set of sensitivity experiments in the new Section 6.

[Figure]

Fig.CR1-1 (as. Fig.6 but with SAL): Difference $d_k$ resulting from the total error of the $M_2$ tide in ICON-O with the SAL parameterization in the R2B6 (top), R2B8 (middle) and BCT configuration (bottom). $d_k$ is defined in Eq.(1) in the main text and evaluated against TPXO9. Unit is cm.

[Figure]

Fig.RC1-2 (as. Fig.7 but with SAL): Same as Fig.RC1-1 but for the difference $d_{k,AM}$ resulting from errors in amplitude only.

[Figure]

Fig.RC1-3 (as. Fig.8 but with SAL): Same as Fig.RC1-1 but for the difference $d_{k,PH}$ resulting from errors in phase only.

[Figure]

Fig.CR1-4 (as. Fig.6 but with TWD): Difference $d_k$ resulting from the total error of the $M_2$ tide in ICON-O with the TWD parameterization in the R2B6 (top), R2B8 (middle) and BCT configuration (bottom).  $d_k$ is defined in Eq.(1) in the main text and evaluated against TPXO9. Unit is cm.

[Figure]

Fig.RC1-5 (as. Fig.7 but with TWD): Same as Fig.RC1-4 but for the difference $d_{k,AM}$ resulting from errors in amplitude only.

[Figure]

Fig.RC1-6 (as. Fig.8 but with TWD): Same as Fig.RC1-4 but for the difference $d_{k,PH}$ resulting from errors in phase only.

**Reply to RC2**

This paper presents a discussion of the realism of the barotropic tides in ICON-O. The paper focuses on the impacts of horizontal resolution, self-attraction and loading and tidal bottom drag (TBD). The conclusions they find are reasonable and consistent with what I understand. My concern is that the primary findings (that SAL+TBD are important) have been discussed in previous papers that are almost 20 years old, for example

Wave drag - Egbert, G.D., R.D. Ray, and B.G. Bills (2004), Numerical modeling of the global semidiurnal tide in the present day and in the last glacial maximum. Journal of Geophysical Research 109, C03003, doi:10.1029/2003JC001973.

SAL - Gordeev, R.G., B.A. Kagan, and E.V. Polyakov (1977), The effects of loading and self-attraction on global ocean tides: The model and the results of a numerical experiment. Journal of Physical Oceanography 7, 161–170, doi:10.1175/1520-0485(1977)007<0161:TEOLAS>2.0.CO.

I'd suggest adding these references to your paper.

We thank the reviewer for his/her comments. In our reply below we use the notion "topographic wave drag" (or TWD), which replaces the notion "tidal bottom drag" (or TWD) following the suggestion of the first reviewer.

The reviewer is right that both SAL and TWD have been discussed in previous papers that are almost 20 years old. However, most of the previous studies, not only the two papers listed above but also the two papers (Pal et al. 2023, Barton et al. 2022) based on the newly developed MPAS-Ocean listed in RC1, consider SAL and TWD in a single-layer tidal model. Generally, parameterizations of SAL and TWD developed for a tidal model that produces only barotropic tides cannot be implemented without any change in a more realistic baroclinic OGCM that produces both tidal and non-tidal motions.

TWD is a drag acting on tidal velocity only. Thus, when implementing TWD in a realistic OGCM, a procedure that identifies tidal velocity from the full velocity should be included. Similar issue is encountered when considering SAL, a secondary potential commonly assumed to be a function of sea level height. Even though SAL can arise from all barotropic motions including non-tidal ones, we assume, following Shihora et al. (2022), that SAL arises mainly from high-frequency barotropic motions and there is a quasi-instantaneous adaptation of the sea water to the time-varying external gravity field for low-frequency motions. Thus, when implementing SAL in a realistic OGCM, one needs to separate sea level height variations arising from high-frequency barotropic motions from the other irrelevant sea level height variations and only use the former in the SAL parameterization.

Arbic et al. (2010) recognized the need of including these extra procedures when implementing parameterizations of SAL and TWD in a multi-layer OGCM. However, it has not been completely clear, which procedures are the best. Here, inspired by Arbic et al. (2010), we applied the simplest method in ICON-O by only considering the full velocity field. To estimate the error for TWD from this simplification, we compared offline from the outputted data the energy removed by TWD equipped with full bottom velocity with the energy removed by TWD equipped with tidal velocity only. We conclude that the error arising by using full (instead of tidal) velocity is negligible. For SAL, we explicitly removed the sea level variations arising from the large-scale time-mean circulation and from baroclinic motions.

Nevertheless, the reviewer made an important point, which we had not properly addressed in the original submission. We now discuss this in the new section 6. We explicitly point out in this section that when implementing SAL and TWD previously developed by Egbert et al. and Godeev et al. for a tidal model in an OGCM, there is a need to identify the relevant parts of velocity and sea level height from the full velocity and full sea level height simulated by the OGCM, and only use these parts of velocity and sea level height in the parameterizations of TWD and SAL. We also point out that further work is needed for finding out the best way for such an identification.

Also, you used a scalar SAL, but state of the art models that I know of use an inline calculation or read in a SAL calculated externally. Scalar SAL is the least accurate of these approaches. I think this point should be mentioned and what your plans are for this important term.

We now made this point clearer in the revised first paragraph in section 5.2. Given the result of Shihora et al. (2022) that implementing a SAL parameterization into an OGCM does not improve the accuracy of the simulated tides to the same degree as it does when implementing the same SAL parameterization into a barotropic model, we decided to try first the simplest approach in this study to assess the potential of this parameterization.

I am viewing this paper as not necessarily containing new science, but rather documenting your efforts to insert tides into a model supporting science in the future. I'd like to see expanded discussion about ICON-O. Some possible discussion points are: What are design choices that make (or will make) ICON-O a better model for investigating internal/external tides compared to other models in the community? What are planned updates for ICON-O? Where does ICON-O have the most issues realistically representing the tides and what does that tell you about deficiencies that need to be investigated/addressed? You state that tides in ICON-O aren't as skillful as HYCOM or MPIOM, what are some of the possible reasons why?

In the new section 6, we have now addressed possible research directions, which can lead to more accurate simulation of tides by ICON-O. For this purpose, we performed a new set of sensitivity experiments, which can partially explain the low

skill of ICON-O.  It seems that the sensitivity to the model time step and the parameters in the time stepping scheme is stronger than what we had anticipated. Certainly, more investigations are needed to fully understand why ICON-O is less skillful than HYCOM and MPIOM.

If these points are addressed I believe the paper will be strengthened and more impactful, because readers will potentially learn information that can help them in their science.

Some specific comments

Line 138 – "Hydrographic" is misspelled

corrected

Line 153 – "Foreman" is misspelled.

corrected

Line 444 – "Gregoris" is misspelled.

corrected

I look forward to reading the revised manuscript..

**Reply to CEC1**

Unfortunately, after checking your manuscript, it has come to our attention that it does not comply with our "Code and Data Policy".
https://www.geoscientific-model-development.net/policies/code_and_data_policy.html

We apologize for not having studied "Code and Data Policy" of GMD carefully and hope that the update is now conform with the journal's policy.

The first problem with your manuscript is that you have not published the code of ICON-O. Our policy clearly states that all the code and data used to produce a manuscript must be publicly available at the submission time in one of the repositories acceptable according to our policy. Actually, your manuscript should not have been accepted in Discussions, given this lack of compliance with our policy. Therefore, the current situation with your manuscript is irregular.

The "Code availability" section has now been amended to comply with the "Code and Data Policy" of GMD.   The new formulation reads:

**Simulations were done with ICON-O version icon-2.6.6. This source code is available on Edmond – the Open Research Data Repository of the Max Planck Society under:**

**https://doi.org/10.17617/3.VQB9N8 (von Storch et al. 2023). The ICON model is available to individuals under the licenses provided under the above link. By downloading the ICON source code, the user accepts the license agreement.**

The "Data availability" section is also wrong. There you cite several repositories, listed later in the References.  First, this is not how this has to be done. You must include the repositories directly in this section, not cite and then list them in the references. Second, the DKRZ is not a repository that complies with our policy, and this should be clear. However, I understand that the 5 TB of data listed are necessary to reproduce your work. In such a case, we can make an exception, as it would be challenging to copy and list such datasets in the regular repositories we list and usually use.

The „Data availability" section has now been amended to comply with the the "Code and Data Policy" of GMD.  DKRZ is of course not a repository, but WDCC (hosted and maintained by DKRZ) is a certified as a Trustworthy Data Repository by CoreTrustSeal (https://www.coretrustseal.org) and complies with your policy. The main data to reproduce the results are published at WDCC. Some supplementary data (like spin-up simulations and raw model output) are published at DOKU at DKRZ. DOKU is the long-term archive at DKRZ and also complies with your policy, even though the data is published without DOIs. PIDs are assigned to the data upon publication.

The new formulation reads:

**The tidal simulations presented in this paper are published at the long-term archive (DOKU) at DKRZ:**

[http://hdl.handle.net/21.14106/a2c5432b7cb7f3a16d126dea36267d01d2abb3d6](http://hdl.handle.net/21.14106/a2c5432b7cb7f3a16d126dea36267d01d2abb3d6) **(Hertwig et al. 2021)**

**These simulations were started from states at the end of the respective spinup simulations published also at the long-term archive (DOKU) at DKRZ:**

[http://hdl.handle.net/21.14106/46e641035ea657a2f90f7ebe6501d327643d1087](http://hdl.handle.net/21.14106/46e641035ea657a2f90f7ebe6501d327643d1087) **(Hertwig et al. 2021)**

**The results of the harmonic analysis (amplitudes and phases of 8 tidal constituents) are published at the long-term archive (DOKU) at DKRZ:**

[https://doi.org/10.26050/WDCC/Tides_ICON-O](https://doi.org/10.26050/WDCC/Tides_ICON-O) **(Hertwig et al.2022)**

**The scripts used for the analysis are published on Zenodo:**

[https://doi.org/10.5281/zenodo.8085145](https://doi.org/10.5281/zenodo.8085145) **(Hertwig & von Storch, 2023)**

**and the TPXO9 data used for ICON-O tides evaluation on Zenodo:**

[https://doi.org/10.5281/zenodo.8074917](https://doi.org/10.5281/zenodo.8074917) **(Hertwig & von Storch, 2023)**

It is important to note that the link to the repository listed as "Hertwig et al. (2022)" is not working. Please, fix it, and reply to this comment with the details for the new one.

The link is fixed in the reference section (an underscore was missing). The correct link is:

[https://doi.org/10.26050/WDCC/Tides_ICON-O](https://doi.org/10.26050/WDCC/Tides_ICON-O)

Also, you say in your manuscript that you use the TPXO9 data to validate your simulations. You must share the data/software that corresponds to the product you have used in the data repository for your manuscript. The TPXO license lets you do it, and the only thing you need to do is include alongside the data the copyright notice that is available on its webpage.

We have published the TPXO9 data used in our manuscript at Zenodo and included it in the „Data availability" section.

Also, currently in the "Code availability" section, you have linked an institutional repository (mpg.de) containing the scripts you used to produce this manuscript. It is laudable that you have done it; however, the mpg.de servers are not an acceptable repository for the long-term archival of assets for scientific publication. Therefore, please, move them to one of the acceptable repositories.

Moreover, in the mpg.de repository containing the scripts, there is no license listed. If you do not include a license, the code is not free-libre-open-source; it continues to be your property, and nobody can use it. Therefore, when uploading the model's code to Zenodo, you could choose a free software/open-source (FLOSS) license. We recommend the GPLv3. You only need to include the file 'https://www.gnu.org/licenses/gpl-3.0.txt' as LICENSE.txt with your

code. Also, you can choose other options that Zenodo provides: GPLv2, Apache License, MIT License, etc.

The scripts are now published at Zenodo (with a license). This is listed under the "Data availability" section.

Finally, you have submitted to our journal your manuscript as a "Model Evaluation Paper". Our author guidelines request that this kind of paper include in the title of the manuscript the version number for the model used. You have not done it. Therefore, in the reply to this comment and potentially reviewed versions of your manuscript, include the version number.

We have included the version number in the title.

Therefore, please, publish your code and data in one of the appropriate repositories, and reply to this comment with the relevant information (link and DOI) as soon as possible, as it must be available for the Discussions stage. Again, please, remember to fix all these issues and to include this information in potentially reviewed versions of your manuscript.

If you do not fix this problem, we will have to reject your manuscript for publication in our journal.

Juan A. Añel

Geosci. Model Dev. Executive Editor